# When Worse is Better: Navigating the Compression-Generation Trade-off for Visual Tokenization

**Vivek Ramanujan**[†‡]  **Kushal Tirumala**[◇]

**Armen Aghajanyan**[‡]  **Luke Zettlemoyer**[◇†]  **Ali Farhadi**[†]

[†]University of Washington  [◇]Meta FAIR
[‡] Work done while at Meta FAIR
ramanv@cs.washington.edu

## Abstract

Current image generation methods are based on a two-stage training approach. In stage 1, an auto-encoder is trained to compress an image into a latent space; in stage 2, a generative model is trained to learn a distribution over that latent space. This reveals a fundamental trade-off, do we compress more aggressively to make the latent distribution easier for the stage 2 model to learn even if it makes reconstruction worse? We study this problem in the context of discrete, auto-regressive image generation. Through the lens of scaling laws, we show that smaller stage 2 models can benefit from more compressed stage 1 latents even if reconstruction performance worsens, demonstrating that generation modeling capacity plays a role in this trade-off. Diving deeper, we rigorously study the connection between compute scaling and the stage 1 rate-distortion trade-off. Next, we introduce Causally Regularized Tokenization (CRT), which uses knowledge of the stage 2 generation modeling procedure to embed useful inductive biases in stage 1 latents. This regularization improves stage 2 generation performance better by making the tokens easier to model without affecting the stage 1 compression rate and marginally affecting distortion: we are able to improve compute efficiency 2-3× over baseline. Finally, we use CRT with further optimizations to the visual tokenizer setup to result in a generative pipeline that matches LlamaGen-3B generation performance (2.18 FID) with half the tokens per image (256 vs. 576) and a fourth the total model parameters (775M vs. 3.1B) while using the same architecture and inference procedure.

## 1 Introduction

Modern image generation methods use a two-stage approach to training. In the first stage, a model such as a VQGAN [13, 63, 46] is trained to compress images to a latent representation. In the second stage, a generative model is trained on these latent representations. This setup implies a deep interaction between the stage 1 and stage 2 models. Since the stage 1 model does not have to reconstruct the image perfectly, it can compress the image more aggressively to achieve a simpler latent distribution for the stage 2 model to learn. However, we ultimately care about the quality of the generated images, which depends on both the stage 1 model's ability to reconstruct and the stage 2 model's ability to learn the latent distribution. On one extreme, if the stage 1 model compresses to a constant, the resulting latent distribution is easy to model but can only generate one image. At the other extreme, the stage 1 model does not compress at all and we get no benefit from latent distribution modeling.

39th Conference on Neural Information Processing Systems (NeurIPS 2025).

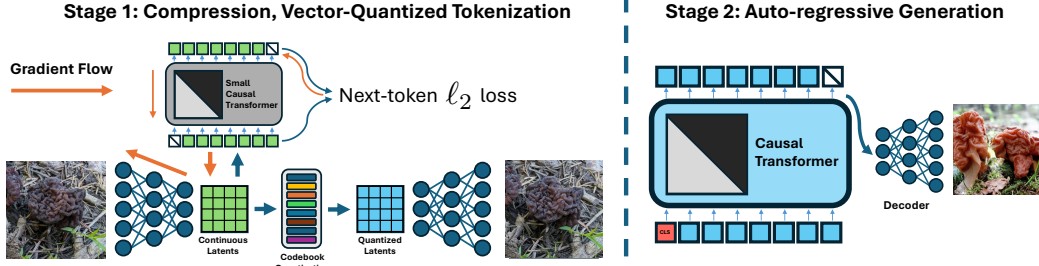

Figure 1: **Overview of our method.** We introduce a small 2-layer causal transformer [41, 51, 11] trained to optimize $\ell_2$ next-token prediction on the pre-quantized latents of the auto-encoder. This loss is propagated through the encoder, producing tokens with a causal transformer inductive bias. Thus, we call our method Causally Regularized Tokenization (CRT).

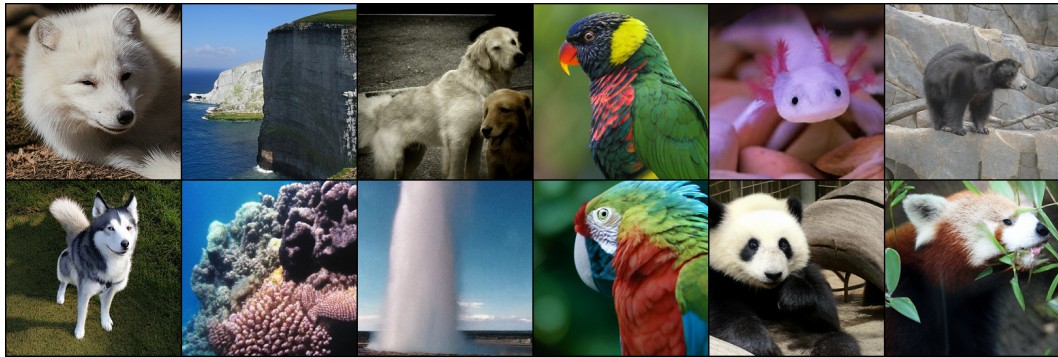

Figure 2: **Qualitative examples from our best model (2.18 gFID).** We show our model's ability to generate high-quality, diverse images on par with LlamaGen-3B [49] with only 775M parameters and $<50\%$ of the tokens per image. For more non-curated examples see Figure 14 and Appendix G.

We view this as a trade-off between rate, distortion, and distribution modeling, similar to the rate-distortion-usefulness trade-off discussed in [52]. The important interacting variables for generation performance are: **(1)** The reconstruction performance of the stage 1 auto-encoder (its "distortion"), **(2)** The degree of compression of the stage 1 auto-encoder (its "rate") and **(3)** The smallest achievable loss of the stage 2 model in learning a distribution over this compressed representation (the "irreducible loss" of distribution modeling).

We study these factors using a vector-quantized GAN (VQGAN) stage 1 model and an auto-regressive stage 2 model trained on VQ tokens. We explore key interventions which affect the stage 1 rate (bits per image pixel) and examine how they affect stage 2 generation performance as a function of model scale. Through the lens of compute scaling laws, we show that compression is not all you need — while small models benefit from more compressed representations, and this relationship also depends on stage 2 training compute. Given this, we ask: *can we design an optimal stage 1 tokenizer given that our stage 2 model is auto-regressive?* Through a simple model-based causal loss, we learn a causal inductive bias on our stage 1 tokenizer. Intuitively, our loss attempts to make token $i$ as predictable as possible given tokens 0 through $i-1$. We call this training recipe Causally Regularized Tokenization (CRT). This regularization trades off reconstruction for distribution modeling, improving generation performance across ImageNet [47] and LSUN [61]. With CRT, we also show significant improvements in generation performance scaling laws (2-2.5x faster training). We demonstrate our improvements through comprehensive experiments across five orders of magnitude of training compute and two orders of magnitude of model scale.

**Summary of contributions.**
· We study the complex trade-off between compression and generation. Our analysis shows that the ideal amount of image compression varies with generation model capacity.
· We provide a principled framework for analyzing this trade-off through the lens of scaling laws, showing consistent patterns across multiple orders of magnitude in computational budget.
· We introduce a method for training a stage 1 tokenizer with a causal inductive bias. This improves inference and training compute scaling of our stage 2 models, without any other interventions, lead-

ing to our key result: by making tokens easier to model, we improve compute efficiency 2-3x over baseline. We also match LlamaGen-3B [49], a prior SOTA auto-regressive discrete generation model (2.18 FID) with half the tokens per image (256 vs 576) and a fourth the total model parameters (775M vs. 3.1B).

## 2 Related Work

For an extensive rundown of visual tokenization methods, we refer the reader to Section I.

**Trade-offs between stage 1 and stage 2 performance.**: We are not the first to notice a disconnect between stage 1 and stage 2 performance. [52] extensively studies how different regularization methods affect the usefulness of learned VAE representations for downstream representations, introducing the "rate-distortion-usefulness" trade-off. Notably, they highlight that stage 1 loss (e.g. VAE loss) is not representative of downstream task usefulness. [53] also shows in Figure 4 demonstrates that increasing $\beta$ in their $\beta$-VAE [23] architecture worsens reconstruction but improves generation performance. Recent advances in lookup-free quantization [37, 64] achieve better codebook utilization, improving stage 1 performance. Both works observe that scaling up codebook size improves stage 1 performance but degrades stage 2 performance (see Figure 3 in [37] and Figure 1 in [64]). Outside visual tokenization, previous work in text tokenization observes that optimizing text tokenizer compression (i.e. stage 1 performance) leads to worse perplexity [33] and downstream accuracy [48]. JetFormer [54] is an end-to-end training procedure for generation, using an auto-regressive (AR) model in conjunction with a flow-based model to perform generation and compression at the same time. They balance this trade-off by using the flow-based model for compression, which by its invertible nature, does not allow for information collapse.

**1D-tokenizers for Generative Visual Models.** Many recent works have identified the issue that existing visual tokenizers are not built for auto-regressive generation and seek to rectify this issue through the use of so-called "1D" tokenizers, which focus on global over local semantics. [17] constructs a discrete tokenizer that captures high-level semantics by using Stable Diffusion [46] as a stage 2 model. VAR and VQVAE–2 [50, 43] modifies the tokenizer to produce multi-scale tokens for easier auto-regressive modeling. [66] and [56] use a transformer register-based [7] approach to learn global tokens with perceptual losses. SEED [16] constructs a 1D tokenizer by using Stable Diffusion [46] as a powerful decoder on quantized tokens. More recently, methods [2, 60, 38] extend this strategy to create progressive tokens, which can be used to flexibly refine generated images for sampling efficiency. In contrast to these works, we do not modify the common VQGAN tokenizer architecture and focus on constructing a causal regularization that enforces compatibility with a stage 2 auto-regressive model. This makes our work generally applicable, as variants of this architecture are common across modern generation systems. We further rigorously study scaling properties with respect to this regularization and other common interventions on tokenizer compression rate, which is not found in existing literature.

## 3 Experiments & Methodology

**Structure of our study.** Our goal is to understand the variables involved in tokenizer construction affecting generation performance at different model scales. First, we fix a stage 1 tokenizer, and study the connection between generation performance (gFID on ImageNet) and compute scaling. We study how changing the stage 1 "rate" (bits per output pixel) affect stages 2 model performance at various scales. Then we examine a new angle on the rate-distortion which affects compute-optimal generation performance scaling: causally regularized tokenization (CRT).

### 3.1 VQGAN and Auto-regressive Image Generation

We summarize the standard auto-regressive image generation procedure [63, 49, 13].

**Stage 1 (Tokenization).** In this stage, the goal is to map an image $x \in \mathbb{R}^{H \times W \times 3}$ to a set of $N$ *discrete* tokens $X_i \in \mathcal{C}$, where $\mathcal{C}$ is a codebook such that $\mathcal{C} \subset \mathbb{R}^d$. We also learn a decoder which can reconstruct the original image from a set of tokens. We use the VQGAN architecture [13], which is a ResNet-based architecture that is a mix of convolutions and self-attention layers in both the encoder and decoder. During training, the encoder maps the input image $x$ to continuous latents

$\hat{X}_i$. Then, we use a codebook look-up to determine the nearest embedding $X_i \in \mathcal{C}$. Finally, these are passed to the decoder to get a reconstruction.

**Stage 2 (Generation).** In this stage, the goal is to learn a prior over the latent tokens learned in the previous stage. For each image in our train set, we use the encoder learned in stage 1 to map it to a set of discrete tokens. We fix an order with which to decode these tokens per image (raster-scan works best empirically [13]), and treat these as sequences. Finally, we train a transformer with causal attention to learn the conditional distributions $p(x_i|x_0 \dots x_{i-1})$ for all tokens $x_i$ by maximizing $\prod_{i=0}^{N} q_\theta(x_i|x_0 \dots x_{i-1})$ for our model $q_\theta$. This is done by minimizing cross-entropy loss over the next-token prediction objective. We train our generation models in the class-conditional regime. To do this, we append a learned class token to the start of the sequence. We also use classifier-free guidance (CFG) at inference time, where we predict a token based on both the class-conditional and unconditional distributions. These decisions are discussed in detail in Section 3.6. Details on architecture, training, and CFG discussed in Appendix A.

**Data.** We train and evaluate primarily on ImageNet (1.28M images, 1000 classes) for class-conditional generation without introducing external data for both stage 1 and stage 2. To confirm general applicability of our method to large scale datasets, we also evaluate our results on several categories of the Large-Scale Scene Understanding Dataset. These categories are cats (1.66M images), horses (2.0M images), and bedrooms (3.02M images).

## 3.2   Metrics

**Reconstruction metrics.** We use Fréchet Inception Distance (FID) [22] on a validation set as our primary metric. We refer to this metric as rFID (reconstruction FID). We also report the signal-to-noise ratio (PSNR) and multi-scale structured similarity metric (MS-SSIM) [59].

**Generation metrics.** We use FID between the generations and the whole validation set for our primary metric. For clarity, we refer to this as generation FID (gFID). To conform with existing literature, we use the OpenAI guided diffusion [10] repository to evaluate our models. We also report Inception Score, precision, and recall, which are proxies for generation fidelity and diversity.

## 3.3   Validation Loss and Generation Performance

### 3.3.1   Neural Scaling Laws for Image Generation

**Scaling Laws for Generation Performance.** We want to study the relationship between validation loss scaling laws (as studied in [29] and [20]) and generation performance scaling as measured by FID. For modeling scaling laws, we use the common [29, 25] functional form $L(C) = L_{min} + L'(C)$ where $L_{min}$ is considered the *irreducible loss* of the problem and $L'$ is the *reducible loss* as a function of $C$ (compute). In our case, we fit $L'(C) = C^\alpha e^\lambda$. When graphed on a log-log scale, $\alpha$ is the slope and $\lambda$ is the $y$-intercept of a line. For both gFID and validation loss scaling laws, we fit $L_{min}$ empirically. We note that rFID is usually equivalent to $L_{min}$, however this is not always the case in practice (see Figure 6 **(far left)**).

**Experimental Setup.** We fix a stage 1 tokenizer (16k token codebook size, 256 sequence length), vary stage 2 model parameters and compute dedicated to training. Fixing codebook size is important because, all else being equal, validation cross-entropy loss is proportional to the intrinsic entropy of the codes, which increases log-log-linearly by codebook size (see Figure 9). Each point in these scaling law plots is an *independent training run*. We only plot the end point of each training run since our cosine learning rate decay schedule results in substantial validation loss changes in the last few iterations of training. Given that we are iterating on a fixed-size dataset ($\sim 3 \times 10^8$ total tokens), we do not scale models past 775M parameters to avoid data scale bottlenecks.

**Log-log-linear loss scaling for visual token modeling.** In Figure 4, we present our findings on the relationship between compute scaling and validation loss. These results corroborate scaling trends found in [20, 29]. We see consistent log-log-linear scaling with validation loss (Figure 4 **(left)**) on the compute validation loss Pareto curve across four orders of magnitude.

**Validation loss and generation performance.** In Figure 4 **(center)**, we plot the gFID score as a function of training compute for different model scales. We see that, in general, lower validation loss implies better generation performance (lower gFID). However, if we look at the upper Pareto frontier

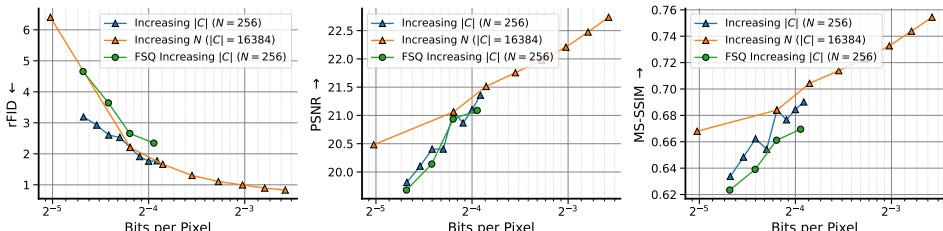

Figure 3: **Relationship between different methods of scaling bits/pixel and reconstruction performance.** Here we increase bits per pixel of either vary the codebook size $|C| \in \{2^{10} \dots 2^{17}\}$ with fixed number of tokens $N = 256$ or by varying $N \in \{196 \dots 576\}$ with fixed $|C| = 2^{14}$ using the strategy described in Section 3.4. Increasing token count is generally more effective at the cost of increased inference compute. We also compare our VQ recipe with FSQ [37], and find it generally outperforms, so we use VQ for all of our experiments going forward.

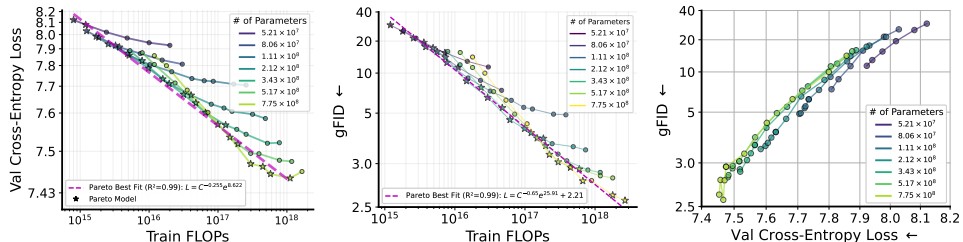

Figure 4: **Neural scaling laws for auto-regressive image generation (ImageNet 256x256)**. We show validation cross-entropy loss (**left**) and gFID (**center**) as a function of total train FLOPs. Across model scales, we observe log-log linear scaling with train FLOPs for both FID and cross-entropy loss across Pareto frontiers. Moreover, we observe validation cross entropy loss is generally predictive of gFID (**right**). See Section 3.3.1 for further discussion.

for Figure 4 (**right**), we see that individual model scales can achieve lower gFID for correspondingly higher validation losses. This is considered the "overtraining" regime with respect to validation loss, described in [15]. This implies that *validation loss is only predictive of FID ordering when the number of training steps is held constant*. More generally, models on the training compute pareto frontier of validation loss are also on the pareto frontier of gFID. This means that *compute optimality with respect to validation loss also corresponds to compute optimality with respect to gFID*, even if specific orderings are not respected between the two quantities.

## 3.4 Sequence Length and Compute Scaling

In this section, we investigate how scaling the tokens per image affects rFID and gFID.

**Setup.** To vary the number of tokens per image, we follow [49] to re-use our tokenizers trained at $256 \times 256$ resolution and evaluate them at higher resolutions. Our base tokenizers are trained with a downsample factor of 16: if we process an $R \times R$ image, the number of tokens produced is $(R/16)^2 = R^2/256$. To measure rFID and PSNR, we take the $R \times R$ output and downsize to $256 \times 256$ with bicubic interpolation. In Figure 3 we observe the expected result that increasing sequence length improves reconstruction performance. It also outperforms increasing codebook size, however at the expense of a linear increase in inference cost. We use this method as opposed to changing the number of encoder downsample layers (e.g. [13, 69, 32]) to avoid introducing confounding variables related to architecture and to provide more granular control over token counts.

**Sequence Length and Scaling Laws.** In Figure 5, we see that for smaller models, training with a lower number of tokens per image is more compute efficient, whereas larger models take advantage of the greater representation capacity of longer sequence lengths. Notably in those results, we find that increasing sequence lengths makes the compute optimal Pareto curve better only close to the 256 token/image rFID lower bound. Our main conclusion is that *more compressed sequences are generally more compute optimal before performance saturation*. As gFID approaches the rFID barrier, using more tokens per image becomes compute optimal.

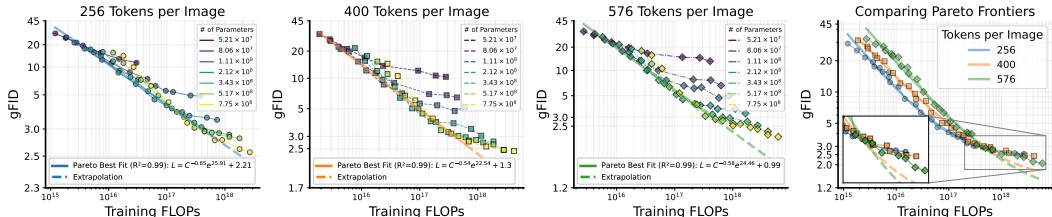

Figure 5: **Relationship between compute scaling laws and tokens per image.** We show compute scaling laws for stage 2 performance for different stage 1 tokens/image. We use the same base tokenizer, expanded to greater tokens/image counts using the strategy described in Section 3.4. We see that fewer tokens per image is generally more compute efficient until we get close to saturation (rFID lower bound), despite their much better reconstruction performance. Also, **(far right)** shows that the true trend deviates more from the predicted trend as tokens per image count increases close to saturation, implying larger models are needed to maintain compute optimality.

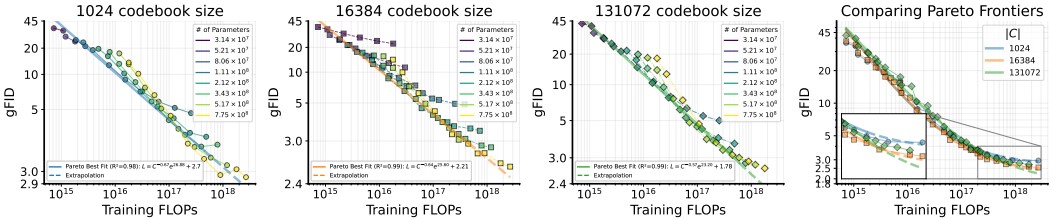

Figure 6: **Relationship between compute scaling laws and stage 1 codebook size.** We show individual model scaling laws at three different stage 1 codebook sizes. 1k outperforms 131k at low compute regimes and the trend reverses at high compute regimes. Note that 16k outperforms both even though it has worse rFID that the 131k codebook size tokenizer (2.21 rFID vs 1.72 rFID).

## 3.5 Codebook Size and Compute Scaling

Codebook size is the second main mechanism for changing the rate of the image tokenizer. This differs from changing sequence length because it has a minor effect on inference cost, both for the stage 1 and stage 2 models, so the effect on scaling is more subtle. This is in contrast to scaling sequence length, which increases inference cost linearly on both stage 2 and stage 1.

**Stage 1 scaling by codebook size.** In Figure 3, we show consistent improvements from 1k to 131k codebook size in stage 1 rFID, PSNR, and MS-SSIM without collapse in performance or codebook utilization, in contrast to prior work [13, 37, 64]. We credit this scaling improvement to modern VQ recipes [62, 49] in conjunction with a long learning rate warm-up period (training recipe details can be found in the appendix). We also show that our VQ recipe outperforms finite-scalar quantization (FSQ) [37], a modern popular non-VQ quantization scheme.

**Stage 2 model scale and codebook size.** In Figure 3, we study scaling laws for the extremes of the codebook sizes we study (1k and 131k), along with our baseline codebook size of 16k. When we use less training compute, we see that the 1k codebook outperforms 131k codebook by 5-10 gFID, even though the rFID is significantly lower. At the other extreme, 131k outperforms 1k, as the stage 2 model nears the gFID saturation point and the worse rFID becomes a bottleneck. This crossover happens around $10^{17}$ train FLOPs. Important to note is that the 16k codebook size outperforms both at almost every compute scale, implying that there is an optimal point for the rFID/gFID trade-off. Interestingly, the change in scaling is not nearly as extreme as that exhibited by the tokens per image ablation in Section 3.4, despite the massive difference in rFID between 1k, 16k and 131k (1.7vs 2.21 vs 3.0 rFID). We suggest that this is because changing codebook size leaves training and inference FLOPs largely unaffected, while compute scales linearly with increased token counts. We discuss this further in Section 4. This also notably differs with prior works [5, 64], which show major differences between stage 2 performance when using VQ with large codebook sizes.

## 3.6 Causally Regularized Tokenization

We have shown how Pareto optimal scaling laws change based on bits per pixel (bpp) used for tokenization. We have also shown that increasing bits per pixel above 16k codebook size and 256

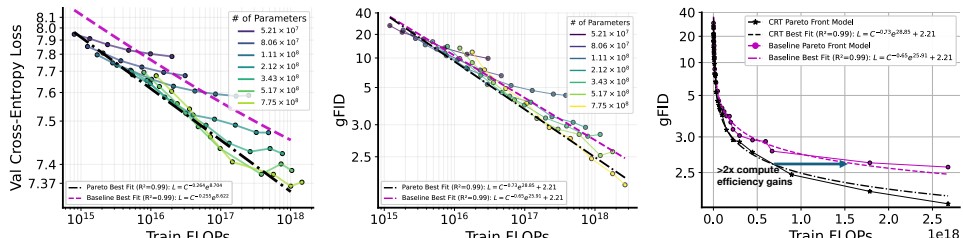

Figure 7: **Stage 2 compute scaling laws with CRT tokens (ImageNet 256x256).** We demonstrate the superior scaling of CRT tokens across four orders of magnitude with respect to validation cross entropy (**left**) and FID (**center**). The trade-off between reconstruction performance and causal dependence means that CRT yields better stage 2 performance across many model scales with worse reconstruction (2.36 vs 2.21 rFID). The result is improved absolute generation performance and $1.5\times$ to $2.5\times$ better scaling by training FLOPs.

Table 1: **System-level and compute-controlled comparisons of autoregressive image-generation models.** (a) *System-level comparison on ImageNet 256×256.* (b) *Compute-controlled comparison against a tuned VQ-GAN baseline on ImageNet.* (c) *Compute-controlled comparison against a tuned VQ-GAN baseline across LSUN Cats, Horses, and Bedrooms.* Across all settings our tokenizer (CRT) and its reconstruction performance optimized variant, *CRT*opt, outperform baselines.

(a) **System-level comparison between methods on ImageNet 256×256**: Our method outperforms other AR, discrete methods at equal parameter counts ($8\times$ less inference compute than LlamaGen-3B for equal performance). *CRT*opt denotes our reconstruction optimized tokenizer variant.

| Type | Model | #Para | Tok/Im↓ | FID↓ | IS↑ | Pre↑ | Rec↑ |
|---|---|---|---|---|---|---|---|
| Diff. | ADM [10] | 554M | – | 10.94 | 101.0 | 0.69 | 0.63 |
| Diff. | LDM-4-G [46] | 400M | – | 3.60 | 247.7 | – | – |
| Diff. | DiT-L/2 [39] | 458M | – | 5.02 | 167.2 | 0.75 | 0.57 |
| Diff. | DiT-XL/2 [39] | 675M | – | 2.27 | 278.2 | 0.83 | 0.57 |
| Diff. | MAR-H [35] | 943M | – | 1.55 | 303.7 | 0.81 | 0.62 |
| Mask. | MaskGIT [3] | 227M | 256 | 6.18 | 182.1 | 0.80 | 0.51 |
| Mask. | RCG (cond.) [34] | 502M | 256 | 3.49 | 215.5 | – | – |
| VAR | VAR-$d16$ [50] | 310M | 680 | 3.30 | 274.4 | 0.84 | 0.51 |
| VAR | VAR-$d20$ [50] | 600M | 680 | 2.57 | 302.6 | 0.83 | 0.56 |
| VAR | VAR-$d24$ [50] | 1.0B | 680 | 2.09 | 312.9 | 0.82 | 0.59 |
| VAR | VAR-$d30$ [50] | 2.0B | 680 | 1.92 | 323.1 | 0.82 | 0.59 |
| AR | VQVAE-2 [43] | 13.5B | 5120 | 31.11 | ∼45 | 0.36 | 0.57 |
| AR | VQGAN [13] | 227M | 256 | 18.65 | 80.4 | 0.78 | 0.26 |
| AR | VQGAN [13] | 1.4B | 256 | 15.78 | 74.3 | – | – |
| AR | ViTVQ [62] | 1.7B | 1024 | 4.17 | 175.1 | – | – |
| AR | LlamaGen-B [49] | 111M | 256 | 6.10 | 182.54 | 0.85 | 0.42 |
| AR | LlamaGen-L [49] | 343M | 256 | 3.80 | 248.28 | 0.83 | 0.51 |
| AR | LlamaGen-XL [49] | 775M | 256 | 3.39 | 227.08 | 0.81 | 0.54 |
| AR | LlamaGen-XXL [49] | 1.4B | 256 | 3.09 | 253.61 | 0.83 | 0.53 |
| AR | LlamaGen-B [49] | 111M | 576 | 5.46 | 193.61 | 0.83 | 0.45 |
| AR | LlamaGen-L [49] | 343M | 576 | 3.07 | 256.06 | 0.83 | 0.52 |
| AR | LlamaGen-XL [49] | 775M | 576 | 2.62 | 244.08 | 0.80 | 0.57 |
| AR | LlamaGen-XXL [49] | 1.4B | 576 | 2.34 | 253.90 | 0.80 | 0.59 |
| AR | LlamaGen-3B [49] | 3.1B | 576 | 2.18 | 263.33 | 0.81 | 0.58 |
| AR | CRT-AR-111M | 111M | 256 | 4.34 | 195.33 | 0.81 | 0.52 |
| AR | CRT-AR-340M | 340M | 256 | 2.75 | 265.24 | 0.83 | 0.54 |
| AR | CRT-AR-775M | 775M | 256 | 2.35 | 259.12 | 0.81 | 0.59 |
| AR | CRT$_{opt}$-AR-111M | 111M | 256 | 4.23 | 194.87 | 0.84 | 0.49 |
| AR | CRT$_{opt}$-AR-340M | 340M | 256 | 2.45 | 249.08 | 0.82 | 0.57 |
| AR | CRT$_{opt}$-AR-775M | 775M | 256 | 2.18 | 268.38 | 0.82 | 0.58 |
| | (validation data) | | | *1.78* | *236.9* | *0.75* | *0.67* |

(b) **Comparison against tuned VQGAN baseline (ImageNet) at saturation.** CRT uniformly out-performs the baseline

| Tok. Type | #Params | Tok/Im | gFID↓ | IS↑ | Prec↑ | Rec↑ |
|---|---|---|---|---|---|---|
| Baseline (2.21 rFID) | 111M | 256 | 4.90 | 180.65 | 0.82 | 0.51 |
| | 211M | 256 | 3.32 | 251.93 | 0.84 | 0.52 |
| | 340M | 256 | 2.89 | 253.25 | 0.83 | 0.54 |
| | 550M | 256 | 2.77 | 277.53 | 0.83 | 0.56 |
| | 775M | 256 | 2.55 | 242.23 | 0.80 | 0.60 |
| **CRT (ours)** (2.36 rFID) | 111M | 256 | 4.34 | 195.33 | 0.81 | 0.52 |
| | 211M | 256 | 2.94 | 241.95 | 0.83 | 0.55 |
| | 340M | 256 | 2.75 | 265.24 | 0.83 | 0.54 |
| | 550M | 256 | 2.55 | 278.29 | 0.80 | 0.59 |
| | 775M | 256 | 2.35 | 259.12 | 0.81 | 0.59 |

(c) **Comparison against tuned VQ-GAN across LSUN datasets**: Under the same compute constraints as (b), CRT uniformly surpasses the tuned baseline on LSUN Cats, Horses, and Bedrooms. gFID$_{CLIP}$ is the primary metric because Inception features are out-of-distribution for LSUN.

| Dataset | Tok. Type | #Params | Tok/Im | gFID$_{clip}$↓ | gFID↓ | Prec↑ | Rec↑ |
|---|---|---|---|---|---|---|---|
| LSUN Cats | Baseline (1.35 rFID) | 111M | 256 | 8.14 | 5.77 | 0.58 | 0.56 |
| | | 340M | 256 | 7.37 | 5.21 | 0.58 | 0.58 |
| | | 775M | 256 | 6.72 | 5.31 | 0.59 | 0.59 |
| | **CRT (ours)** (1.48 rFID) | 111M | 256 | 7.22 | 5.32 | 0.60 | 0.54 |
| | | 340M | 256 | 6.77 | 4.98 | 0.61 | 0.56 |
| | | 775M | 256 | 6.74 | 5.32 | 0.60 | 0.57 |
| LSUN Horses | Baseline (1.65 rFID) | 111M | 256 | 5.56 | 3.19 | 0.62 | 0.62 |
| | | 340M | 256 | 5.09 | 3.00 | 0.62 | 0.62 |
| | | 775M | 256 | 4.92 | 2.56 | 0.62 | 0.63 |
| | **CRT (ours)** (1.83 rFID) | 111M | 256 | 5.20 | 2.65 | 0.61 | 0.61 |
| | | 340M | 256 | 4.63 | 2.62 | 0.61 | 0.63 |
| | | 775M | 256 | 4.05 | 2.76 | 0.61 | 0.63 |
| LSUN Bedrooms | Baseline (0.99 rFID) | 111M | 256 | 9.81 | 2.57 | 0.59 | 0.53 |
| | | 340M | 256 | 9.61 | 2.51 | 0.58 | 0.55 |
| | | 775M | 256 | 9.50 | 2.53 | 0.57 | 0.57 |
| | **CRT (ours)** (1.39 rFID) | 111M | 256 | 9.06 | 2.37 | 0.60 | 0.53 |
| | | 340M | 256 | 8.56 | 2.08 | 0.59 | 0.56 |
| | | 775M | 256 | 8.50 | 2.22 | 0.58 | 0.57 |

tokens per image generally makes scaling worse. Further, while reducing to 1k codebook size improves the scaling law marginally, explicitly reducing the reconstruction capacity to such a degree greatly harms generation performance close to saturation. Going past bpp as a measure of rate, we study an alternate rate-distortion trade-off based on optimizing stage 1 latents for stage 2 performance. Our guiding question is: *since our stage 2 model is an auto-regressive transformer, can we imbue this inductive bias into the tokenizer?* We show we can take our optimal baseline tokenizer and apply a simple causal regularization that improves scaling without changing bpp used for compression.

**Setup.** Figure 1 demonstrates our key intervention. We apply an $\ell_2$ loss parameterized by a causal transformer to the tokens *before* quantization, and backpropagate this loss to the encoder of the our stage 1 auto-encoder. We train with this model-based loss ($\mathcal{L}_{CRT}$) in conjunction standard VQGAN

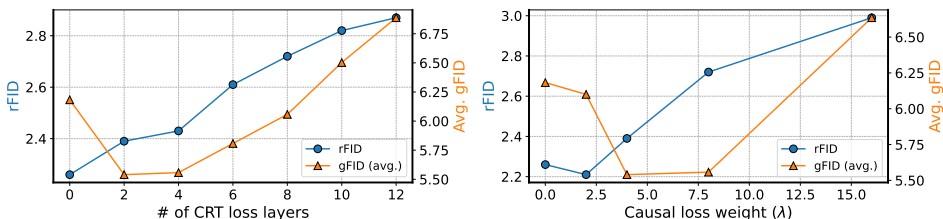

Figure 8: **Impact of CRT hyperparameter choices on reconstruction FID (rFID) and generation FID (gFID). (top)**: Increasing the number of causal regularizer layers worsens rFID and, beyond 4 layers, also degrades gFID. **(bottom)**: Higher causal loss weights ($\lambda$) trade off rFID for improved next-token prediction, with optimal balance at $\lambda = 4$. 0 layers and $\lambda = 0$ correspond to baseline. Further discussion in Section 3.6.

losses, using the same hyperparameters as our baseline recipe. Applying this loss with cross-entropy after quantization is intuitive; however, we found that this damages reconstruction fidelity too much (see Section 4), since VQ tokens change discretely during training. If token $i$ changes to token $j$, the cross-entropy loss treats it the same regardless of how similar the tokens $i$ and $j$ are. Meanwhile, $\ell_2$ on pre-quantized tokens are naturally similarity aware. We use a 2-layer causal transformer with the architecture of our stage 2 model. This introduces a 5% cost in training FLOPs, which we control for by training 5% less (2 fewer epochs). For both our baseline and CRT models we use a codebook of size 16k and 256 tokens per image.

**Effects on stage 1 reconstruction.** After applying this loss, our tokenizer is marginally worse in terms of reconstruction against the compute normalized baseline (2.36 rFID vs 2.21 rFID). Intuitively, our loss attempts to make token $i$ as predictable as possible given tokens 0 through $i - 1$. The most predictable set of tokens would be if the encoder outputs a sequence of entirely dependent tokens (e.g., the constant token). However, this reduces the total information that can be conveyed through the tokens. Our CRT loss (a proxy for generation) conflicts with reconstruction, implying a trade-off. With this inductive bias, stage 2 scaling is improved, which we discuss next.

**Improvements in stage 2 performance.** In Figure 7, we show the relative scaling properties of stage 2 models using our CRT tokenizer compared to the VQ-GAN baseline (in Figure 4). For clarity, we also include Pareto frontier scaling trends from Figure 4. Figure 7 **(left)** Shows that stage 2 models trained with our tokenizer achieve systematically lower validation losses. Figure 7 **(center)** shows that this translates to improvements generation performance on the training compute Pareto front. Namely, our compute scaling law is *improved* from $\alpha = -0.65$ to $\alpha = -0.73$. Finally, Figure 7 **(right)** shows the actual magnitude of improvement, which the log-log-linear plots obfuscate: CRT achieves gFID scores with $1.5 - 3\times$ compute efficiency and improve over the baseline, even with worse rFID. We show comparisons at individual model scales in Table 1b, and detailed comparisons across training durations in Appendix F. Our model systematically outperforms the baseline in all compute and model size regimes.

**Results on LSUN** [61]. In Table 1c, demonstrate our method's generality by evaluating several LSUN categories. We demonstrate improvements in gFID and gFID$_{CLIP}$ across three model scales using CRT across all model scales. Note that we use CLIP gFID because these datasets are out of distribution for the Inception network, making FID a more noisy metric.

**Optimizing the CRT tokenizer for absolute performance.** In the previous sections, we perform compute-controlled comparisons of CRT and baseline. CRT outperforms the baseline despite the trade-off in reconstruction performance. However, we showed previously that generation performance close to the rFID barrier is affected by reconstruction quality. In this section, we apply simple optimizations to boost reconstruction performance, which do not modify our fundamental setup, to push the limits of our method. These include: (1) increasing training duration from 38 epochs to 80 epochs, (2) doubling the size of the decoder, (3) increasing the codebook size from 16k to 131k, which interestingly improves performance for CRT but not our baseline. With these changes, our tokenizer achieves **1.26 rFID**. Our stage 2 training procedures remain constant. In Table 1a, we show a system-level comparison of our method (CRT$_{opt}$-AR) to other existing methods in the field. The closest comparison to our method is LlamaGen, which uses the same architectures and inference method. Notably, we exceed the performance of the LlamaGen 3.1B parameter model with our 775M parameter model, while using half the number of tokens per image. This is

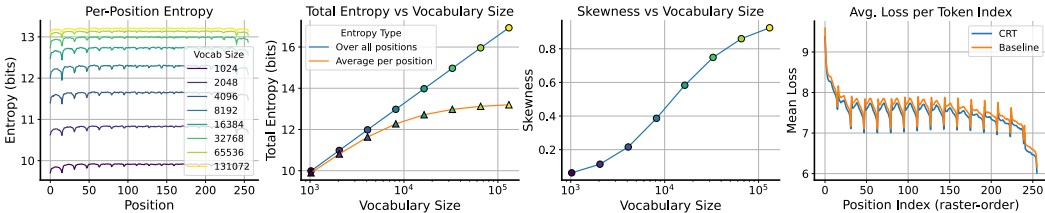

Figure 9: **Entropy analysis of image tokenizers.** We do a deep analysis of how tokenizer entropy distributions. Per position and total entropy deviate with increasing codebook size, implying $n$-gram modeling gains which are not reflected in codebook utilization. Skewness **(center right)** is another view of the concentration of codes property demonstrated by **(center left)**. We compute skewness as $1 - \frac{2^{\text{entropy per position}}}{2^{\text{total entropy}}}$, demonstrating that as the codebook size increases, so does codebook specialization per position. **(far right)** shows CRT tokenization uniformly reduce stage 2 loss across all positions, improving stage 2 generation performance. See Section 4 for more details.

an **8x reduction** in inference compute. We show uncurated qualitative examples from select object categories in Figure 14 and further examples in Appendix G.

**Hyperparameter Ablations.** There are two key choices involved with our causal loss: **(1)** the size of the causal model used during stage 1 training and **(2)** the weight $\lambda$ we assign to the $\ell_2$ next-token prediction loss ($\lambda \mathcal{L}_{CRT}$). We include an ablation of these parameters in Figure 8. We plot the relationship of rFID and gFID with respect to these parameters, computing gFID over an average of 4 stage 2 model scales (52M, 111M, 211M, 340M) trained for 250k iterations each. In Figure 8 **(left)**, we increase the number of layers in our causal regularizer. For every 2-layers we add, we reduce the overall training time by 5% to compensate. We see that rFID uniformly increases with more layers, as expected. After 4 layers, this worsening rFID hurts average gFID. Thus, we chose to use two layers in all of our experiments, which has the added advantage of being very computationally cheap. In Figure 8 **(right)**, we see a similar trend. Increasing $\lambda$ biases the encoder to focus on making representations easy for next-token prediction rather than reconstruction, making overall rFID worse. However, there's a sweet spot around $\lambda = 4$ where this worsened reconstruction is within acceptable range and this also improves overall generation performance.

**Loss Function Ablation ($\ell_2$ vs cross-entropy).** In Appendix E, we show comparisons between $\ell_2$ loss and cross-entropy (CE) loss. We also compare to the loss suggested by [56], who propose a similar auto-regressive inductive bias in the video domain (based on CE not $\ell_2$). In general, CE losses harm reconstruction performance in our setting but fail to provide substantial generation performance improvement to be worth the trade-off. This deviation from findings in [56] is likely due to two reasons: **(1)** They use an architecture that is substantially different from ours, focusing on global versus local tokens (we use the VQGAN architecture for compatibility with existing SOTA literature) **(2)** The video generation setting they studied is much further from performance saturation, meaning changes to tokenizer reconstruction are less likely to harm generation performance.

## 4 Analysis

**Factors affecting validation loss.** We saw in Section 3.4 that tokens per image affected compute scaling more significantly than changing the codebook size. Why is that? We note that for a stage 2 unigram model, the loss lower bound $L_{min}$ is the empirical entropy of the codebook distribution per-position, since cross-entropy can be decomposed into $H(X) + D_{KL}(q(X)|p(X))$ and $D_{KL} \rightarrow 0$ when $q \rightarrow p$. In Figure 9 **(far left)**, we measure this directly over the ImageNet train set over various codebook sizes. In Figure 9 **(center left)**, we see that the per position entropy *converges*, even as the overall codebook entropy increases log-linearly due to uniform codebook utilization. This implies that as codebook size increases, codes become more specialized by position. This has the effect of ensuring that $L_{min}$ for a unigram model does not increase, meaning that our stage 2 model can still learn even with very large codebook sizes.

**What does CRT do?** $L_{min}$ for an $n$-gram model is upper-bounded by this estimate, as $H(X_i) \geq H(X_i|X_{<i})$. $H(X_i|X_{<i})$ is not directly measurable, this quantity is what we are trying to estimate with language modeling. We directly model and minimize this in stage 1 with CRT, using the same architecture as in stage 2, to ensure that our inductive biases transfer. In Figure 9 **(far right)**, we plot the stage 2 losses per-position for a 211M parameter auto-regressive model between our baseline

tokenizer and CRT. These tokenizers have the same codebook size, utilization, and per-position entropy. However, we see that CRT lowers loss uniformly across token positions, with most of the loss reduction happening at the end of the sequence. This implies that the CRT tokenizer has less estimated entropy than baseline when using a causal entropy model.

## 5 Conclusion & Future Work

We systematically study the tradeoff between compression and generation for image tokenization and reveal several key insights: First, we demonstrate that the relationship between compression and generation quality is nuanced: smaller models benefit from more aggressive compression, even at the cost of reconstruction fidelity. Second, we provide a principled framework for analyzing this trade-off through the lens of scaling laws, showing consistent patterns across multiple orders of magnitude in computational budget. Finally, we introduce Causally Regularized Tokenization (CRT), a method for optimizing tokenizers specifically for autoregressive generation. We add a causal inductive biases during tokenizer training which substantially improves compute efficiency (2-3x) and parameter efficiency (4x reduction) compared to previous approaches. Notably, these gains are most pronounced in the resource-constrained regime, making the method particularly valuable for practical applications. Our results highlight the importance of considering the interaction between different components for multi-stage machine learning pipelines: rather than optimizing each stage independently, we achieve significant gains by baking in inductive biases in early stages. This principle suggests several promising directions for future work, for example extending to other architectures (e.g. diffusion models) or other modalities (e.g. video and audio tokenizers).

## 6 Acknowledgments

The authors thank Jonathan Hayase, Matthew Wallingford, Ludwig Schmidt, Nicholas Lourie, Will Merrill, Mohammand Rastegari, Jaskirat Singh, Lili Yu, Chunting Zhou, Artidoro Pagnoni and members of the RAIVN Lab for helpful comments and discussions throughout the course of research. VR and AF acknowledge funding by NSF IIS 1652052, IIS 1703166, DARPA N66001-19-2-4031, DARPA W911NF-15-1-0543 and gifts from Allen Institute for Artificial Intelligence, Google and Apple.

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

# A  Further training details

In this section, we enumerate the complete training details for experiments in Section 3. These basic hyperparameters are the same for both

## A.1  Stage 1: VQGAN training

We largely follow the training recipes from [13] and [49] to train our tokenizer, with modifications that we state here. The VQGAN architecture is exactly that of [13], with the following standard loss terms (not including our CRT loss):

$$\mathcal{L} = \lambda_{\text{VQ}}\mathcal{L}_{\text{VQ}} + \lambda_{\text{GAN}}\mathcal{L}_{\text{GAN}} + \lambda_{\text{Perceptual}}\mathcal{L}_{\text{Perceptual}} + \lambda_{L2}\mathcal{L}_2. \tag{1}$$

The perceptual loss comes from LPIPS [67]. The $\ell_2$ loss is defined as $\mathcal{L}_2(x, y) = \|x - y\|_2^2$ between the reconstruction and the input. The weights for each loss are: $\lambda_{VQ} = 1.0$, $\lambda_{GAN} = 0.5$, $\lambda_{Perceptual} = 1.0$, $\lambda_{L2} = 1.0$. We start $\lambda_{GAN}$ at 0. After 20k iterations, we anneal $\lambda_{GAN}$ to 0.5 using a cosine schedule for 2k iterations.

**GAN Loss Details.** The GAN loss uses the PatchGAN [27] as a discriminator with the Wasserstein GAN loss function from [1] directly on reconstructions from the auto-encoder to force the output to look "realistic." This is the only "reference-free" image loss used. Using improved StyleGAN and StyleGAN2 [30] discriminators ended up with worse qualitative results in our setting.

**Codebook Loss Details.** Following [62, 49], we project from 256 to 8 dimensions before doing codebook lookup and perform codebook lookup with cosine distance. We use two losses, a quantization error loss and a commitment loss, both proposed in [55], with a commitment $\beta = 0.25$. We do not use entropy loss, as it was unnecessary to prevent codebook collapse in our setup. We find that training with a long learning rate warmup period is crucial for scaling beyond 16k codes, otherwise performance saturates at 2.1 rFID and codebook utilization stagnates.

**Optimization Parameters.** We use Adam [31] with learning rate $1e - 4$ and $\beta = (0.9, 0.95)$. AdamW with weight decay $0.1$ performs identically. We use the same optimization parameters for the PatchGAN discriminator. We use with linear warmup from $1e-5$ for 3k iterations, batch size 128 and a constant learning rate. We found that this warmup is critical for good performance, especially for codebook length and training duration scaling. Cosine learning rate gave very similar results, and constant learning rate allows for continued training. We train for a total of 400k iterations unless stated otherwise.

**CRT details.** CRT is our L2 next-token prediction loss applied to the raster-order tokens during stage 1 training. The weight for this loss for all of our experiments is 4.0 (we include a sweep in Figure 8) Like with the GAN discriminator, we anneal this weight from 0 to 4.0 for 1k iterations from the start of training. The duration of this annealing is not important for final performance. The architecture we use is identical to our stage 2 model, except with no last classification layer and no token embedding layer (replaced with a single linear layer). We use a separate optimizer for this component: an AdamW optimizer with learning rate and $\beta$ parameters matching the original VQGAN at all iterations. Our weight decay was $0.1$. We normalize the loss by the annealed weight at all iterations so that the causal model training is not impacted by its weight. To control for extra training FLOPs, we drop the number of training iterations to 380k. For CRT$_{opt}$, we train for 800k iterations.

Table 2: **Stage 2 model configurations.** These are the parameters for the Llama architectures we use in our stage 2 experiment sweeps. To construct this sweep, we interpolated by head and layer counts with existing standard models (-S, -M, -L, etc.) and ensured 64 dimensions per head.

| Parameters | Heads | Layers | Dimension |
|---|---|---|---|
| $5.21 \times 10^7$ | 10 | 6 | 640 |
| $8.06 \times 10^7$ | 11 | 9 | 704 |
| $1.11 \times 10^8$ | 12 | 12 | 768 |
| $2.12 \times 10^8$ | 14 | 18 | 896 |
| $3.43 \times 10^8$ | 16 | 24 | 1024 |
| $5.17 \times 10^8$ | 18 | 30 | 1152 |
| $7.75 \times 10^8$ | 20 | 36 | 1280 |

## A.2 Stage 2: Auto-regressive training

**Architecture.** Following [49], we use the Llama 2 architecture [51] with QK-layer normalization [8] for stability. For some 576 tokens/image runs, we also needed to use z-loss [6] with a coefficient of $1e - 4$ for stability. We find that this did not impact the performance of stable runs.

**Optimization.** We perform data augmentation using ten-crop, allowing us to pre-tokenize the train set during stage 2 training. We use AdamW, betas $(0.9, 0.95)$ and a cosine learning with schedule with a linear warmup (5k iterations) and a batch size of 256. Our max learning rate is $3e - 3$, with a pre-warmup learning rate of $3e - 4$, annealing our learning rate to 0. We also include a class token dropout rate of 0.1, where we replace the class-token with a randomly initialized learnable dummy token to enable classifier-free guidance. We train for a total of 375k iterations unless otherwise stated (we often sweep this parameter for stage 2 scaling law experiments).

**Class-conditioning and Classifier-Free Guidance (CFG).** For class-conditioning, we prepend a special class token to image token sequences at train time. At inference time, we use the relevant class token to start the sequence to generate an image of a particular class. We sample with next-token prediction without top-$k$ or top-$p$ sampling. All our results use CFG at inference time to further emphasize classifier guidance. To do this, we have two inference procedures, one with the class-token and one using the aforementioned dummy token. At each timestep, these yield logits $\ell_u$ for the unconditional logit, and $\ell_c$ for the class-conditioned logit. To get the next token, we sample from the combined logit $\ell_{cfg} = \ell_u + (\ell_c - \ell_u)\alpha$, where $\alpha$ is known at the CFG scale. This kind of sampling was initially used for diffusion [24], but has seen success in auto-regressive image generation as well [65, 49] and is used commonly.

CFG trades off fidelity and diversity (related to precision and recall respectively) of generation, and we found that the optimal CFG scale was also pareto optimal for precision and recall. To choose our CFG scale at inference time, we split a 50k subset of the train set (not held out during training) to compute FID against. We select the optimal FID against samples from $\alpha \in \{1.5, 1.75, 2.0, 2.25\}$ and use those generations against the validation set. We do this because CFG can have a large effect on FID score [35, 49, 50], and optimizing directly against the ImageNet validation set can pollute our evaluation. However, in separate experiments, we find that optimal CFG against this train split and the validation set are identical in practice.

## A.3 Scaling law sweep details

To produce scaling laws (e.g. Figure 4), we sweep training compute by changing the number of total training iterations. Because we are in the repeated data setting, unlike [29], and we are using a cosine learning rate schedule, we need to evaluate the end points of separate, independent training runs to produce these curves. The iteration counts we sweep over are $\{15k, 22.5k, 45k, 90k, 187.5k, 375k, 750k, 1.5M, 2.25M\}$ for all models. We compute estimates for training FLOPs using the approximation from [29]. The architecture parameters for our sweep can be found in Table 2.

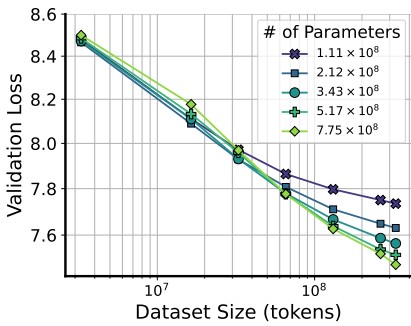

Figure 10: **Stage 2 Validation Loss by Dataset Size.** Here we study the data scale bottleneck by varying the total size of the dataset. Every model was trained for 250k iterations at batch size 256. We see that our dataset is large enough to demonstrate scaling laws at the parameter range studied.

### A.4 Computational Cost

Each experiment can be run on one node of 8 80GB A100s. Training the tokenizer for 40 epochs takes 3 days in this setting. The largest stage 2 model (770M) trained for 375k iterations takes 3.5 days, and the cost decreases linearly with the number of parameters and training iterations.

## B  Limitations

Our work has a few specific limitations, largely related to the availability of compute and data. First, the largest model we test is 775M, which we do in order to prevent overfitting (given that ImageNet is a finite dataset). This is close to the range of current state-of-the-art image generation models, such as [12] which ranges from 800M parameters to 8B parameters, but is smaller than modern large language models [25, 51]. Future work could include scaling parameter counts. Second, we primarily evaluate on class-to-image synthesis and not text-to-image synthesis. We do this because FID is a much more well-behaved metric on ImageNet (see inconsistent scaling in LSUN results by parameter count in Table 1c). However, this means we cannot train on web-scale datasets, and thus are in the repeated data regime, deviating from [29]. In Figure 10, we study the effects of this but varying the stage 2 training set size (375k training iterations), showing that for existing model sizes we are not yet in the data bottleneck regime. Future work could include validating an in distribution metric (e.g. CLIP FID) on a large scale text-to-image dataset and then reproducing our scaling laws.

## C  Broader Impacts

Our method, as applied in this paper, makes training of image generation models more efficient and boosts overall performance. Image generation in general has broader social impacts. For example, Deepfakes can cause personal harm. Popular text-to-image models can be seen as stealing artist data for corporate gain. In its most benign form, this method can be used to accelerate research on fully legal datasets (e.g. Creative Commons images), however that may not be how it is used in the future.

## D  Further Baseline Tokenizer Ablations

Table 3 shows the difference between native resolution training and resolution extrapolation (the method used in the paper). Native resolution training improves performance very little compared to extrapolating to higher resolution images. This allows us to compare a fixed tokenizer at many different compression rates, removing the confounding aspect of higher resolution training.

Figure **??** shows the result of training with and without a long learning rate warmup period. This was crucial for achieving high codebook utilization and good performance without collapse for large codebooks.

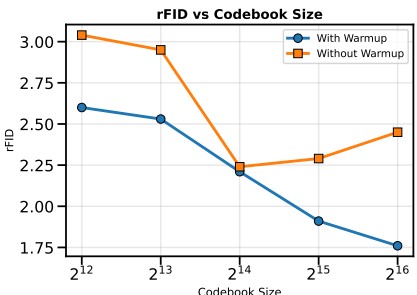

Figure 11: **Effects of learning rate warmup on tokenizer performance.**

Table 3: **Training Tokenizers at Native Resolution.** Native resolution training improves performance very little compared to extrapolating to higher resolution images. This allows us to compare a fixed tokenizer at many different compression rates, removing the confound of higher resolution training.

| Tokens per Image | Native resolution training | rFID / gFID |
|:---:|:---:|:---:|
| 400 | no | 1.30 / 3.76 |
| 400 | yes | 1.21 / 3.72 |
| 576 | no | 0.99 / 3.61 |
| 576 | yes | 0.93 / 3.66 |

## E    Further CRT Ablations

Results for different causal loss application configurations are presented in Table 4.

Table 4: Comparison between $\ell_2$ loss and CE loss for causal regularization. The LARP loss is also compared with components relevant to this setup (CE + stochastic VQ + cosine distance + scheduled sampling [56]). The setup from [56] is likely more successful in conjunction with global register tokens as opposed to patch tokens (as in our case). We report average gFID across stage 2 scales 52M, 111M, 211M, and 340M trained for 250k iterations each.

| Causal Loss Form | rFID | Avg. gFID |
|:---:|:---:|:---:|
| $\ell_2$ before quantization (CRT) | 2.36 | 5.51 |
| $\ell_2$ after quantization | 2.61 | 6.37 |
| CE after quantization ($\lambda = 0.03$) | 2.53 | 6.10 |
| LARP (CE + SVQ + scheduled sampling) | 2.67 | 6.08 |

**Effects of decoding order on performance.** As noted in prior work [13] raster-scan is empirically the best canonical ordering for image generation. Our results corroborate this. While CRT provides improvements in these settings, it is not enough to overcome the difference in ordering performance (>1gFID). It is possible that a more flexible base architecture (e.g. a 1D transformer based tokenizer) than that of VQGAN would allow for more flexible re-ordering, and we believe this to be an interesting direction for future work.

## F    More Detailed Scaling Plots

In this section we cover more detailed scaling law explorations expanding on plots in the main paper.

Figure 13 provides a different perspective on the comparison between Figures 4 and 7. We do this by comparing stage 2 model scales directly by gFID performance by train FLOPs. Note that like the main figure, each point is an independent training run. We see that CRT tokens improve performance at all scales both for training efficiency and final performance. This difference is especially pronounced at the smaller compute and parameter scales.

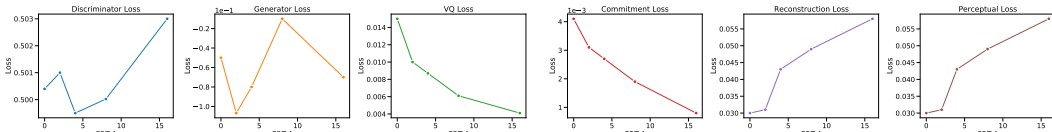

Figure 12: **Effects of CRT weight $\lambda$ on VQGAN losses.** In general, increasing loss weight increases losses related to perceptual quantities (e.g. $\ell_2$ reconstruction, discriminator loss, and perceptual LPIPS loss). It also seems to reduce codebook losses, like VQ and commitment.

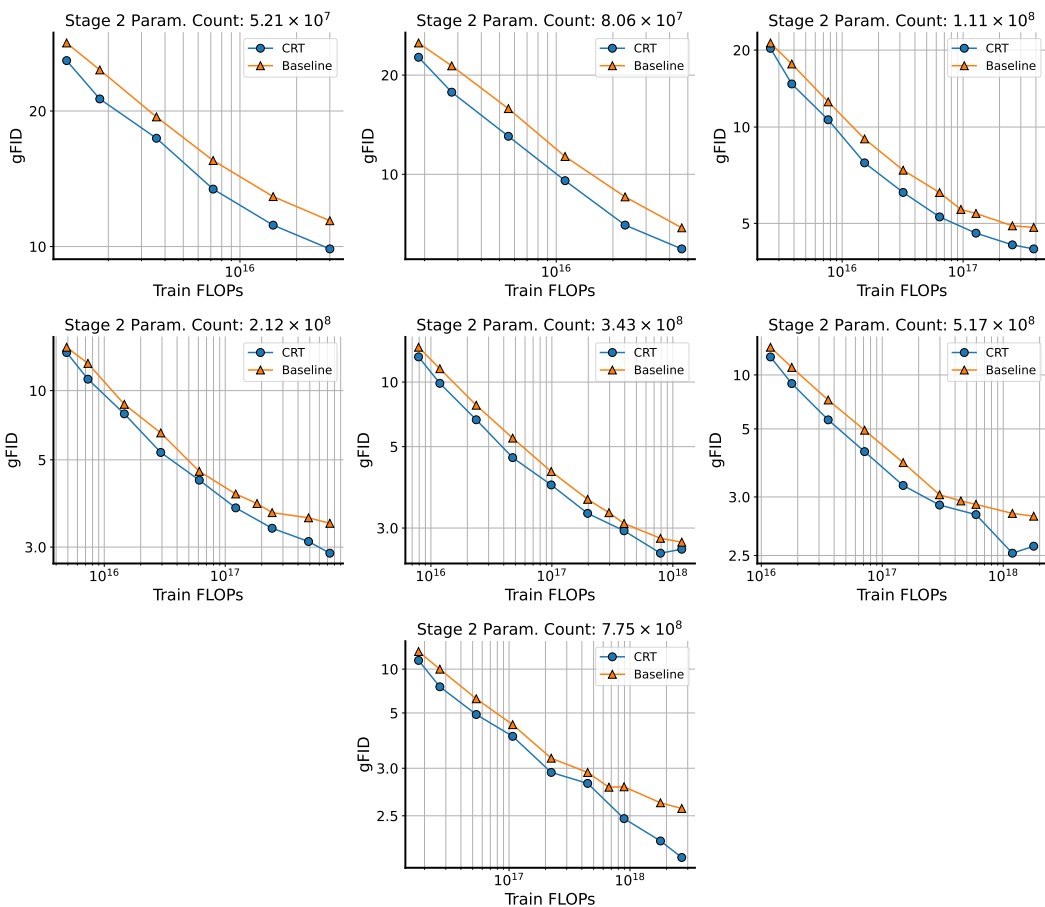

Figure 13: **Individual model scale comparison for Figures 4 and 7.** Here we show how stage 2 gFID scales with respect to train FLOPs with both the CRT and baseline tokenizer. We see that with CRT tokens, stage 2 learns faster and also ends up more performant across all model sizes, especially at smaller scales (note the log-scaled y-axis).

## G   Qualitative Examples

Figures 15-19 show non-cherrypicked generation examples from our best model $CRT_{opt}$-AR-775M. They demonstrate a high degree of diversity and image quality across a wide array of classes. Figure 29 shows selected comparisons with and without CRT, constructed so that overall image structure is comparable. CRT demonstrates better long-range and structural consistency.

## H   Text-to-image results

The main body of this work focused on ImageNet, since it is an established benchmark with good image diversity. However, much of the large-scale work on image generation focuses on text-to-image

Table 5: **Text-to-image results on BLIP-3o.** Re-using the CRT (opt variant) and baseline tokenizers, we compare results on the BLIP-3o pre-training dataset for text-to-image performance (775M auto-regressive model for 600k iterations). We show improvements on both image-text alignment and distribution matching.

| Tokenizer | gFID | CLIPScore |
|-----------|------|-----------|
| Baseline | 4.61 | 0.33 |
| CRT (ours) | 4.15 | 0.36 |

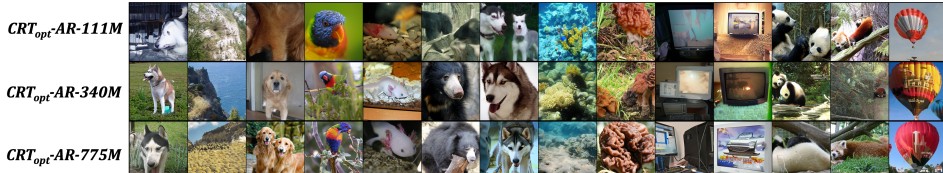

Figure 14: **Qualitative examples (best viewed zoomed in).** Random, non-curated samples from $CRT_{opt}$-AR models of increasing scale (111M to 775M parameters, ImageNet 256x256). Larger models demonstrate improved visual fidelity and coherence across diverse subjects including animals, landscapes, and objects across scales. CRT allows for even small models to produce high quality outputs.

generation. To show the generality of our method, we show results in the text-to-image domain. For this experiment, we took the BLIP-3o long-caption [4] pre-training dataset (27M image-text pairs vs 1.2M ImageNet images), tokenized the text with OpenAI's CLIP L/14 text tower [42], and trained an auto-regressive image generation model conditioned on these text embeddings. This dataset is diverse, being a combination of webcrawled images and high-quality data from JourneyDB. We trained our XL sized model (775M) for 600k iterations at batch size 256 using the CRT (opt variant) and baseline tokenizers from our study. The results from this experiment are in Table 5. We use CLIPScore [21] (higher is better) with OpenAI CLIP B/32 [42] to measure image-text alignment. We see a solid improvement with our method, even though we are re-using the image tokenizer from our ImageNet experiments and this dataset is thus OOD.

# I  Extended Related Work

**Evolution of visual tokenization.** The current paradigm of discrete visual tokenizers came from VQ-VAE [55], originally introduced as a method for discrete latent representation learning in continuous domains such as image and audio. It was introduced to prevent the posterior collapse problem in VAEs by separating the compression and generation phases of training. Section 4.2 of [55] notes that their loss function leads to blurry reconstructions, and the authors suggest including perceptual metrics (metrics that align with human judgment of similarity) in the loss function. [14] explores this idea with VQGAN, which introduces a perceptual loss based on $LPIPS$ [67], and a GAN loss to add realistic textures. Since the advent of VQGAN, there have been numerous attempts at improving visual tokenizers [62, 19, 69, 68, 66, 57, 28, 45, 58, 40, 70, 44]. Some works focus on improvements including ViT-VQGAN [62], which replaces CNN layers with ViT layers, and MoVQGAN [69], which adds adaptive instance normalization layers in the CNN decoder. Other works focus on mitigating "codebook collapse" [26] in vector quantization (VQ) [18] by introducing novel quantization schemes, for example by replacing dead codes with encoder outputs [9]. We refer the reader to [26] for a thorough review of these methods. Other works replace VQ entirely with scalar quantization techniques [37], and demonstrates increased codebook utilization compared to VQ: for example, MAG-VIT [64, 36] quantizes each dimension to 2 values and adds entropy loss to encourage codebook utilization.

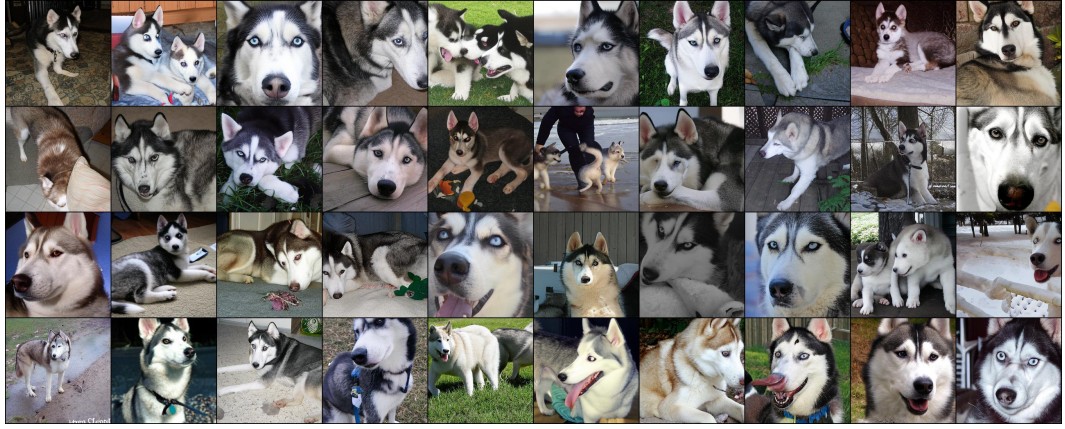

Figure 15: **Non-cherrypicked generation examples from CRT$_{opt}$-AR-775M (ImageNet 256x256).** CFG scale = 4.0, class index 250, class name Siberian Husky

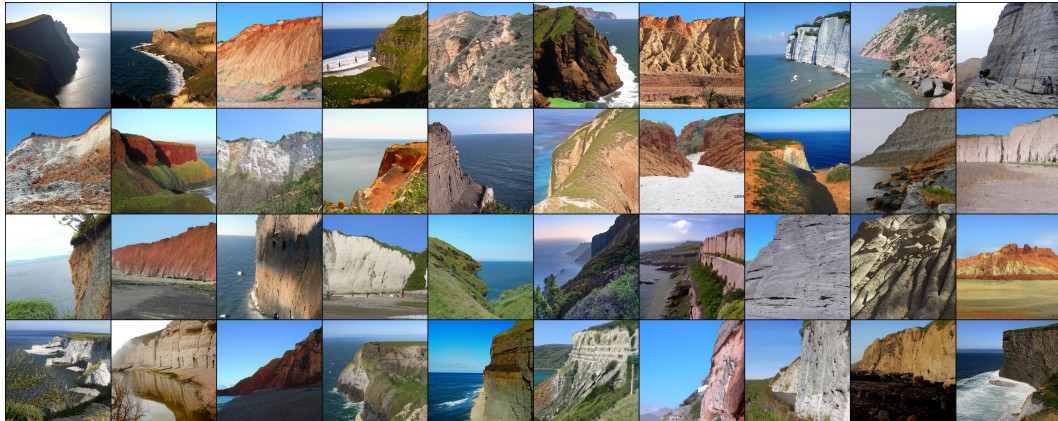

Figure 16: **Non-cherrypicked generation examples from CRT$_{opt}$-AR-775M (ImageNet 256x256).** CFG scale = 4.0, class index 972, class name cliff

## J    VQVAE comparisons

In Figure 30, we show detailed comparisons between reconstructions from baseline and various CRT configurations. CRT and baseline look quite similar, while CRT with CE (instead of L2) demonstrates some noticeable blocky artifacts as a result of the strength of regularization.

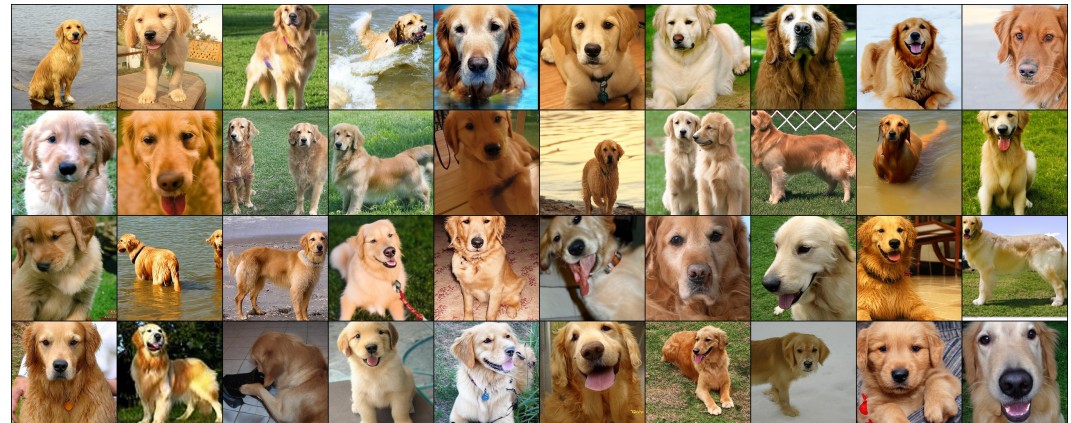

Figure 17: **Non-cherrypicked generation examples from CRT$_{opt}$-AR-775M (ImageNet 256x256).** CFG scale = 4.0, class index 207, class name Golden Retriever

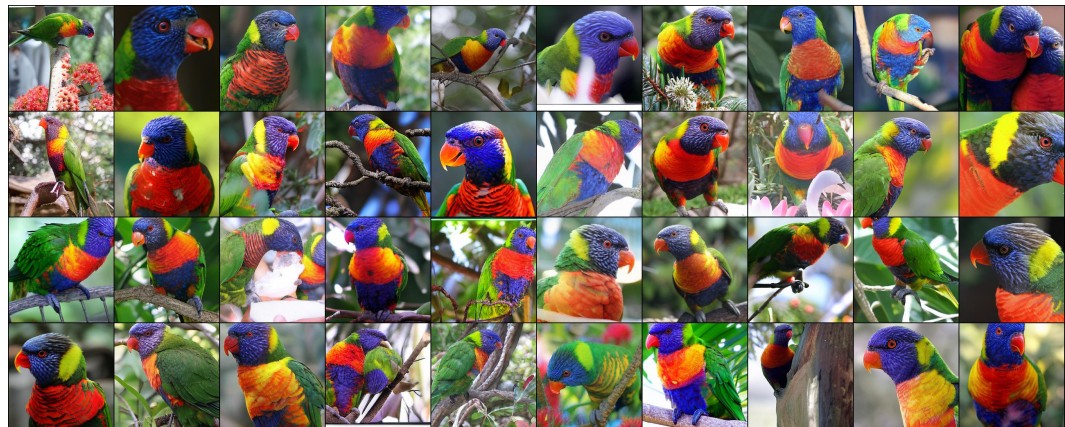

Figure 18: **Non-cherrypicked generation examples from CRT$_{opt}$-AR-775M (ImageNet 256x256).** CFG scale = 4.0, class index 90, class name lorikeet

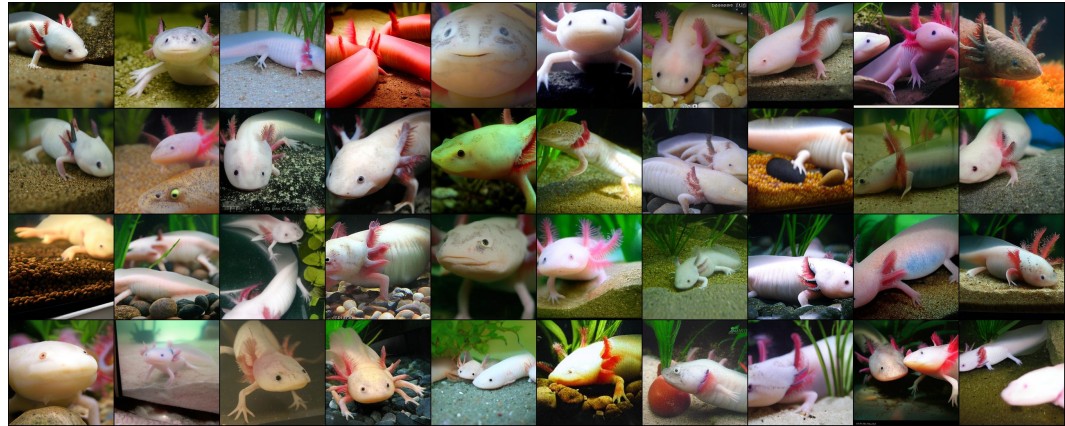

Figure 19: **Non-cherrypicked generation examples from CRT$_{opt}$-AR-775M (ImageNet 256x256).** CFG scale = 4.0, class index 29, class name axolotl

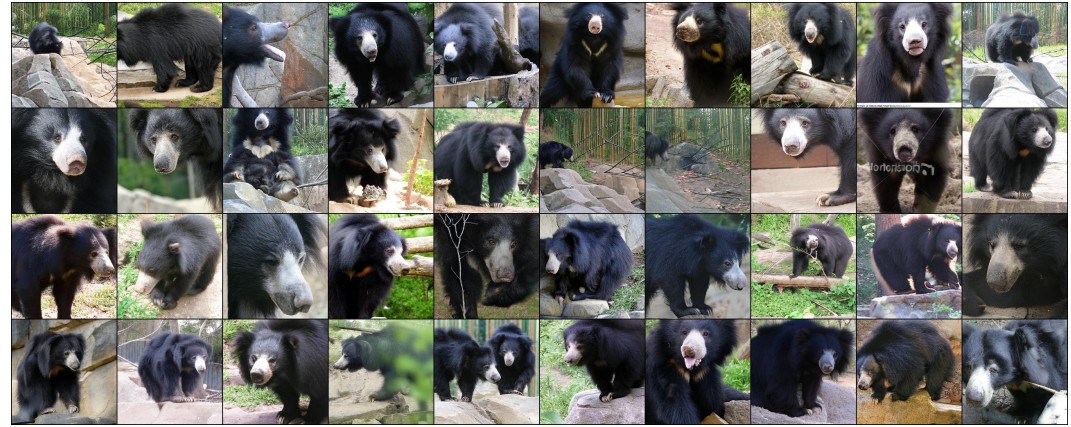

Figure 20: **Non-cherrypicked generation examples from CRT$_{opt}$-AR-775M (ImageNet 256x256).** CFG scale = 4.0, class index 297, class name sloth bear

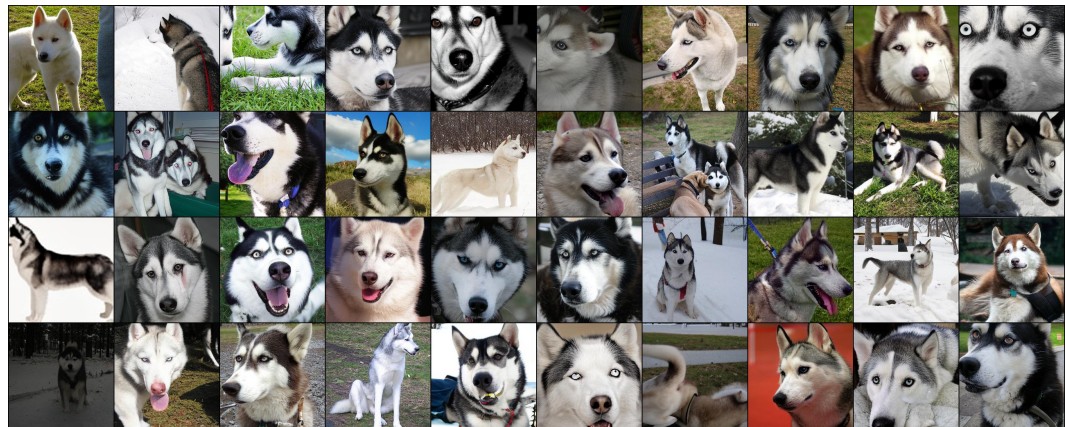

Figure 21: **Non-cherrypicked generation examples from CRT$_{opt}$-AR-775M (ImageNet 256x256).** CFG scale = 4.0, class index 248, class name husky

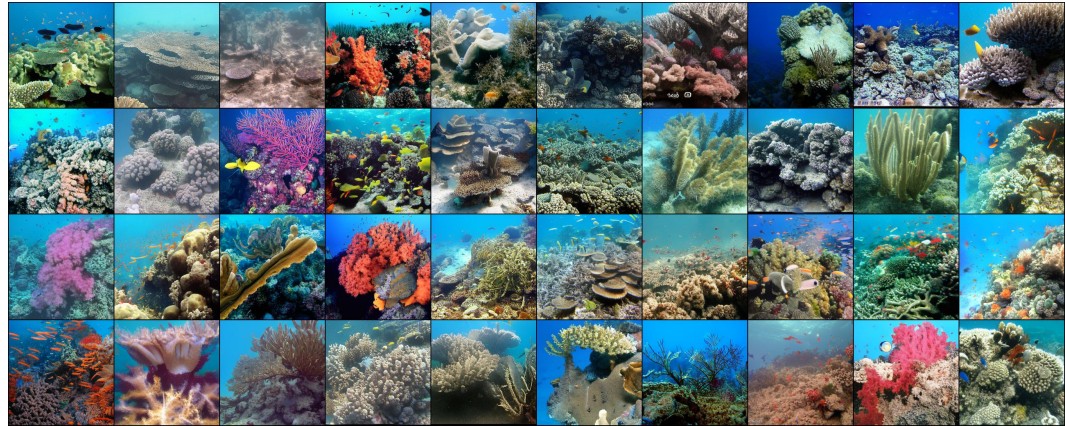

Figure 22: **Non-cherrypicked generation examples from CRT$_{opt}$-AR-775M (ImageNet 256x256).** CFG scale = 4.0, class index 973, class name coral reef

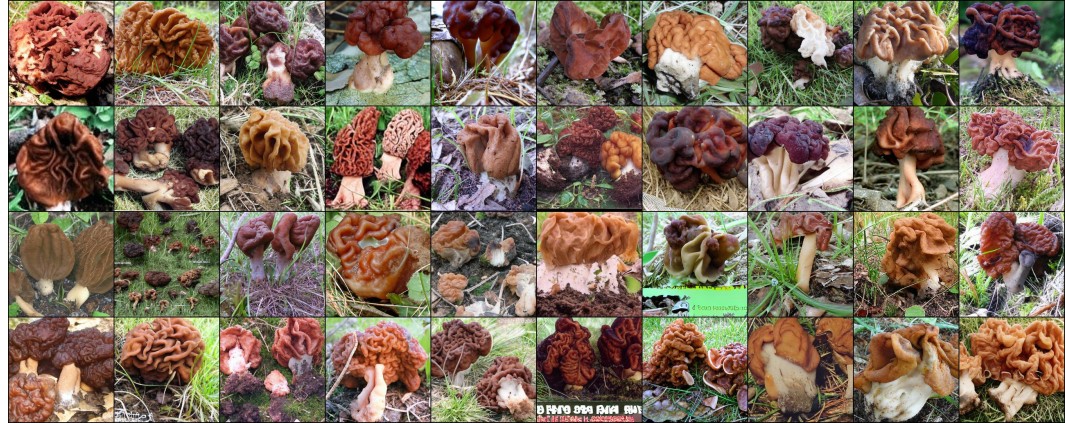

Figure 23: **Non-cherrypicked generation examples from CRT$_{opt}$-AR-775M (ImageNet 256x256).** CFG scale = 4.0, class index 993, class name gyromitra

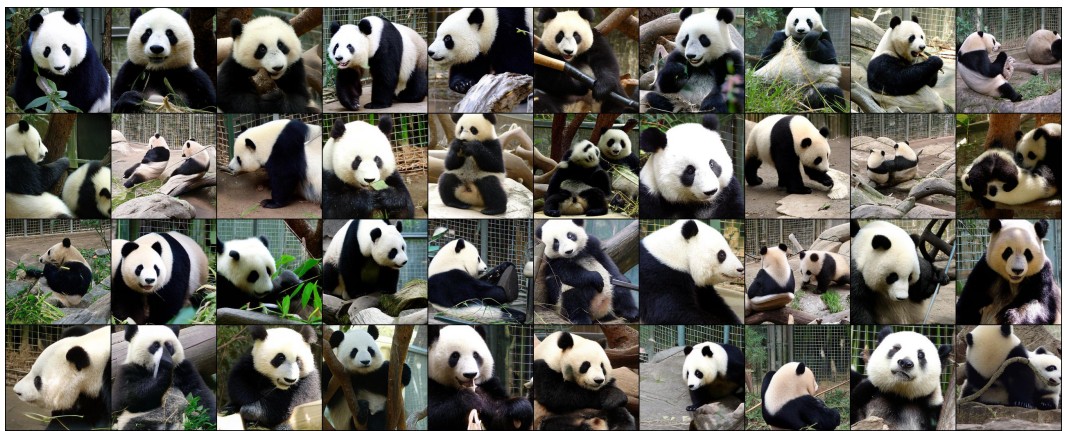

Figure 24: **Non-cherrypicked generation examples from CRT$_{opt}$-AR-775M (ImageNet 256x256).** CFG scale = 4.0, class index 388, class name giant panda

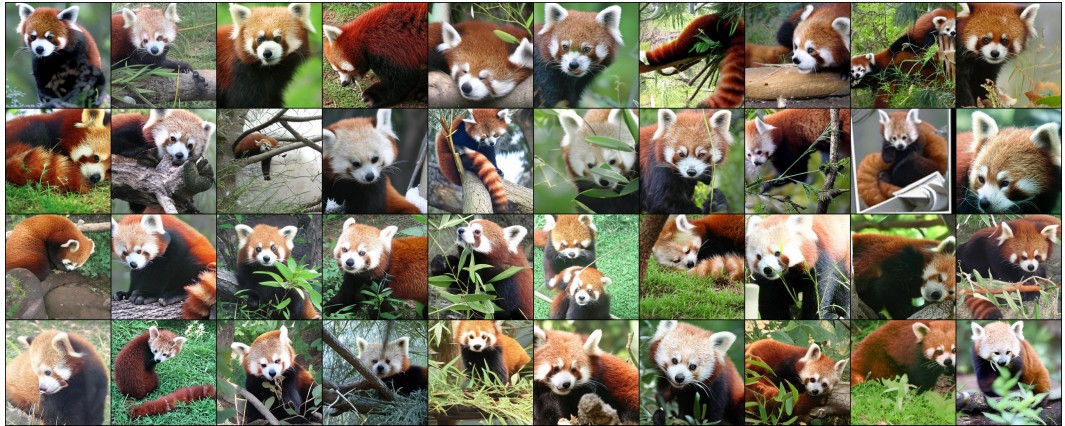

Figure 25: **Non-cherrypicked generation examples from CRT$_{opt}$-AR-775M (ImageNet 256x256).** CFG scale = 4.0, class index 387, class name red panda

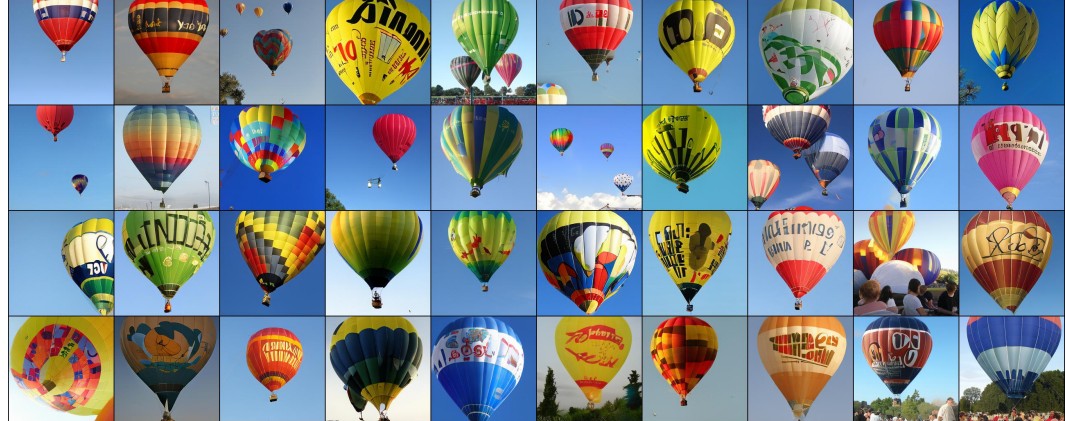

Figure 26: **Non-cherrypicked generation examples from CRT$_{opt}$-AR-775M (ImageNet 256x256).** CFG scale = 4.0, class index 417, class name balloon

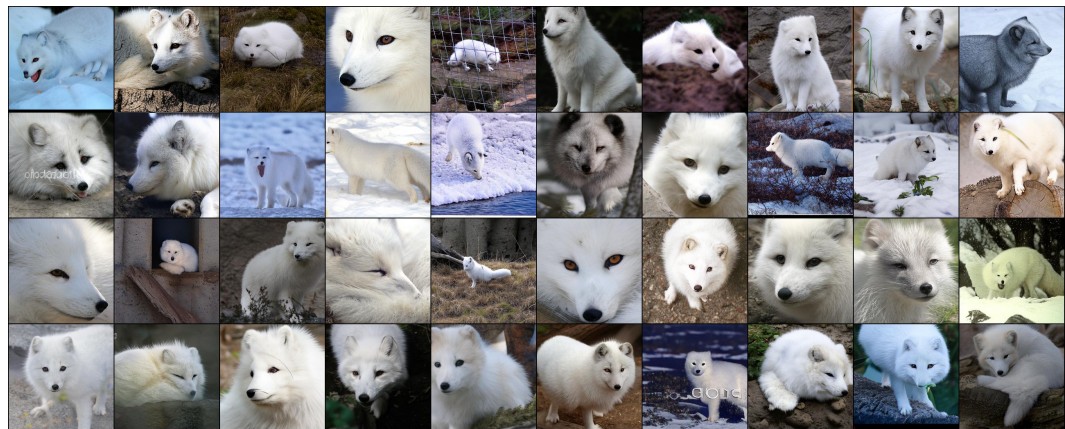

Figure 27: **Non-cherrypicked generation examples from CRT$_{opt}$-AR-775M (ImageNet 256x256).** CFG scale = 4.0, class index 279, class name Arctic fox

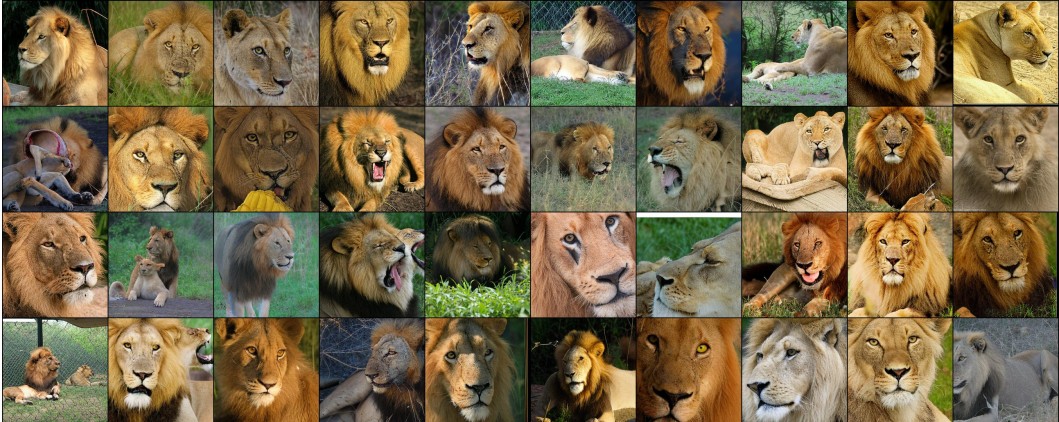

Figure 28: **Non-cherrypicked generation examples from CRT$_{opt}$-AR-775M (ImageNet 256x256).** CFG scale = 4.0, class index 291, class name lion

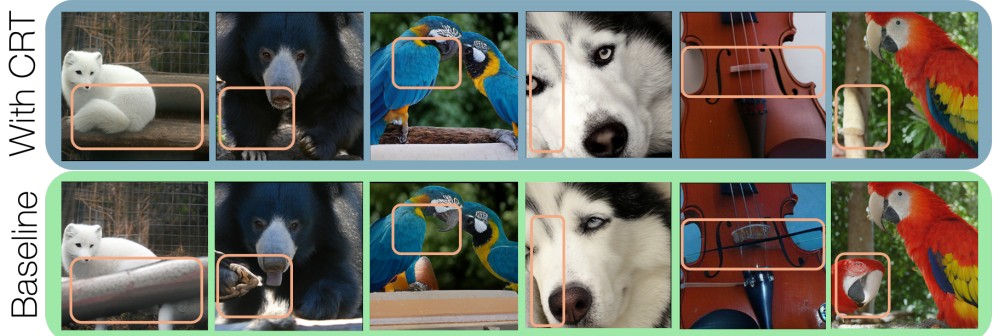

Figure 29: **Selected generation comparisons with and without using the CRT tokenizer.** Here we show direct comparisons between stage 2 generations with and without using the CRT tokenizer. CRT results in greater long-range structural coherence and higher quality samples. To generate these direct comparisons, we provide the first 32 (of 256) tokens from a validation example for context to the stage 2 auto-regressive model and filter for image similarity using CLIP B/32 [42] embeddings. For more non-curated generations with our model, see Figure 14 and Appendix G.

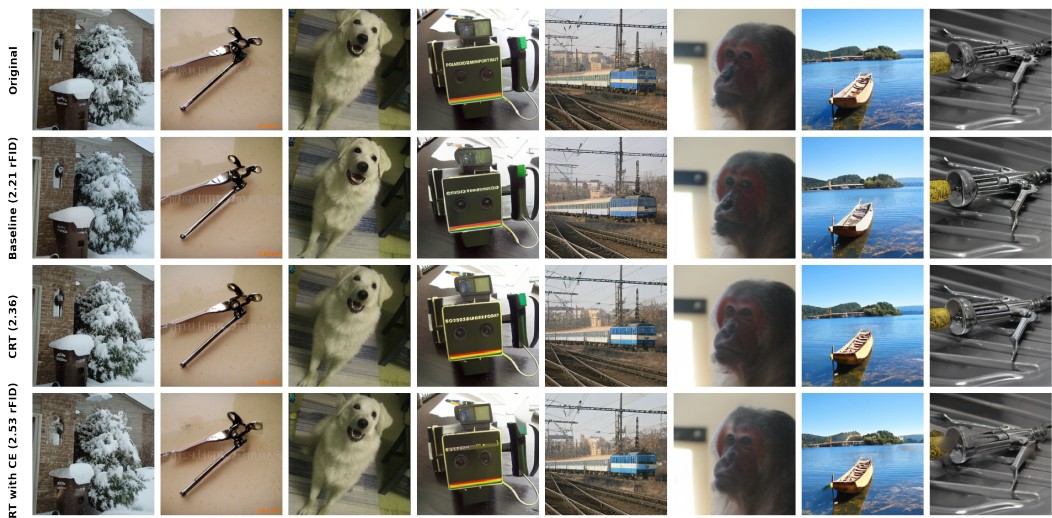

Figure 30: **Reconstruction comparison (best viewed zoomed in).** We show that CRT applied to baseline only minimally affects distortion, while CRT with CE introduces more noticeable blocky artifacts in reconstruction due to the strength of regularization.

