# OpenReview forum: "When Worse is Better: Navigating the Compression Generation Trade-off In Visual Tokenization"
_NeurIPS.cc/2025/Conference — NeurIPS 2025 spotlight_

### Official Review · Reviewer_pH7h · 2025-06-30

**Clarity:** 3
**Significance:** 2
**Originality:** 2
**Rating:** 4
**Confidence:** 4

**Summary:**

In this paper, the authors systematically investigate the trade-off between compression and generation, including neural scaling, sequence length scaling and codebook size scaling. Based on these analysis, the authors shows that the ideal amount of image compression varies with generation model capacity. In addition, they also introduce Causally Regularized Tokenization (CRT) to train tokenizers with a causal inductive bias. CRT improves generation performance across ImageNet and LSUN.

**Questions:**

See Weakness.

**Ethical Concerns:**

["NO or VERY MINOR ethics concerns only"]

**Final Justification:**

My concerns have been resolved, thus, I raise my score to Borderline accept.

**Limitations:**

yes

**Quality:**

2

**Strengths And Weaknesses:**

Strengths
+ This paper provides a systematically investigation about the trade-off between compression and generation. The analysis on sequence length scaling and codebook size scaling is somewhat new.
+ This paper imbues the causal inductive bias of generation model into the tokenizer by  Causally Regularized Tokenization (CRT). This simple causal regularization boost generation performance across various datasets.

Weaknesses
- The link between the systematic investigation and CRT is not very strong, which makes their combination in one paper feel somewhat disjointed.
-  Although this paper provides systematic investigation about the trade-off between compression and generation, these investigations are not absolutely new and insightful. Neural scaling is widely studied in previous works, like VAR. Sequence length scaling and codebook size scaling are implicitly studied in existing works, like LlamaGen.
- Limited experiments about CRT. In this paper, the effectiveness of CRT is only validated on a codebook of size 16k and 256 tokens per image. CRT is designed for image tokenizers. It is essential to provide more experiments to show effectiveness with various codebook size and tokens per image.
- Missing discussion about Figure 9 (center right). In the main text, there is no discussion about skewness vs. vocabulary size.
- Type Errors. 1) line 156, higher -> lower; 2) line 199, (1.7 vs 2.21 vs 3.0 rFID) - > (3.0 vs 2.21 vs 1.7 rFID).

---

> ### Author Rebuttal · Authors · 2025-07-28
>
> ### Addressing Weaknesses
>
> **The link between the systematic investigation and CRT is not very strong, which makes their combination in one paper feel somewhat disjointed.**
>
> The first half of the paper shows that increasing bits per pixel above 16k codebook size and 256 tokens per image generally makes scaling worse. Further, while reducing to 1k codebook size improves the scaling law marginally, explicitly reducing the reconstruction capacity to such a degree greatly harms generation performance close to saturation. This motivates us to look for a rate-distortion trade-off which does not directly inhibit the bottleneck size, of which CRT is one. Section 4 analysis expands on this concept. We will make this explicit in the text, which should improve the overall flow of the paper.
>
> **Although this paper provides systematic investigation about the trade-off between compression and generation, these investigations are not absolutely new and insightful. \[e.g. VAR or LlamaGen\]**
>
> We would like to clarify that the _existence_ of scaling laws in this scenario is not surprising. Scaling laws have been widely studied and validated since Kaplan et al. 2020. The connection between scaling laws and the rate-distortion trade-off (which we measure as bits per pixel) is not studied in prior art. The reviewer points to VAR and LlamaGen. We address our novelty to these individual studies below:
>
>
> **VAR.** VAR includes scaling laws with regard to a proposed generation procedure (next-scale generation) with a specific, fixed stage 1 tokenizer. We study the effect changing the tokenizer's compression rate has on stage 2 scaling, a dimension not explored in VAR.
>
> **LlamaGen.** LlamaGen includes some results with increasing token count and stage 2 model capacity, however, they do not attempt to establish scaling laws and thus miss much of the interplay between compute scale and model capacity. They also do not study the effect of codebook size on scaling. Further, their codebook scaling does not get 100% codebook utilization or uniform codebook distribution, resulting in subpar scaling beyond 16k codebook size. For scaling laws to be valid, components of the pipeline have to be tuned and optimized.
>
> We emphasize that the setup of studying stage 1 rate-distortion in connection to stage 2 compute scaling laws is, to our knowledge, unique in literature. Having factors related to stage 1 bottleneck capacity studied together in a unified, systematic setting is crucial for understanding the interplay of rate-distortion and stage 2 modeling.
>
> **Limited experiments about CRT. In this paper, the effectiveness of CRT is only validated on a codebook of size 16k and 256 tokens per image. CRT is designed for image tokenizers. It is essential to provide more experiments to show effectiveness with various codebook size and tokens per image.**
>
>
> We thank the reviewer for this important point. Given the depth of our scaling law study, we decided to focus CRT on the setting which demonstrated the best trade-off between reconstruction and generation in the first part of our paper (16k codebook size with 256 tokens per image). We show additional results at multiple scales with more tokens per image below (400 and 576). We observe that across tokens per image configurations and stage-2 model size, CRT outperforms the baseline.
>
> **400 tokens per image**
>
> | Parameter Count | Iterations | Method | FID |
> | :--- | :--- | :--- | :--- |
> | **111M** | $7.5 \times 10^5$ | Baseline | 5.3745 |
> | | $7.5 \times 10^5$ | CRT | 4.7140 |
> | | $2.25 \times 10^6$ | Baseline | 5.4305 |
> | | $2.25 \times 10^6$ | CRT | 4.3158 |
> | **211M** | $7.5 \times 10^5$ | Baseline | 3.7601 |
> | | $7.5 \times 10^5$ | CRT | 3.3526 |
> | | $2.25 \times 10^6$ | Baseline | 3.4438 |
> | | $2.25 \times 10^6$ | CRT | 3.0547 |
> | **775M** | $2.25 \times 10^6$ | Baseline | 2.4247 |
> | | $2.25 \times 10^6$ | CRT | 2.2492 |
>
>
> **576 Tokens per Image**
>
> Of course, here is the table with the **Precision** and **Recall** columns removed.
>
> | Parameter Count | Iterations | Method | FID |
> | :--- | :--- | :--- | :--- |
> | **111M** | $7.5 \times 10^5$ | Baseline | 6.2174 |
> | | $7.5 \times 10^5$ | CRT | 4.7268 |
> | | $1.5 \times 10^6$ | Baseline | 4.7259 |
> | | $1.5 \times 10^6$ | CRT | 4.5559 |
> | **211M** | $7.5 \times 10^5$ | Baseline | 3.6164 |
> | | $7.5 \times 10^5$ | CRT | 3.3401 |
> | | $1.5 \times 10^6$ | Baseline | 3.6185 |
> | | $1.5 \times 10^6$ | CRT | 3.1098 |
> | **775M** | $7.5 \times 10^5$ | Baseline | 2.4453 |
> | | $7.5 \times 10^5$ | CRT | 2.1984 |
> | | $1.5 \times 10^6$ | Baseline | 2.2381 |
> | | $1.5 \times 10^6$ | CRT | 2.0993 |
>
>
> We also ran $CRT_{opt}$ in settings with greater codebook size. We show those results below.
>
> | Parameter Count | Codebook Size | Method | FID |
> | :--- | :--- | :--- | :--- |
> | **111M** | 16384 | Baseline | 4.8837 |
> | | 16384 | $CRT_{opt}$ | 4.2323 |
> | | 131072 | Baseline | 5.0036 |
> | | 131072 | $CRT_{opt}$ | 4.4945 |
> | **550M** | 16384 | Baseline | 2.8497 |
> | | 16384 | $CRT_{opt}$ | 2.5793 |
> | | 131072 | Baseline | 2.8454 |
> | | 131072 | $CRT_{opt}$ | 2.4488 |
> | **775M** | 16384 | Baseline | 2.5452 |
> | | 16384 | $CRT_{opt}$ | 2.2079 |
> | | 131072 | Baseline | 2.7080 |
> | | 131072 | $CRT_{opt}$ | 2.1833 |
>
>
> It is computationally infeasible for us to generate full scaling law studies for each specific setting, but we hope the results above are evidence that our conclusion is not specific to a particular tokenizer configuration.
>
>
> **Missing discussion about Figure 9 (center right). In the main text, there is no discussion about skewness vs. vocabulary size.**
>
>
> We thank the reviewer for pointing this out. Skewness is another view of the concentration of codes property demonstrated by Figure 9 (center left). We compute skewness as $1 - \frac{2^\text{entropy per position}}{2^\text{total entropy}}$, demonstrating that as the codebook size increases, so does codebook specialization per position. We will include this discussion in the final revision.
>
>
> If we have adequately responded to the concerns with our experiments, we request that the reviewer raise their score.

---

> > ### Comment · Reviewer_pH7h · 2025-08-05
> >
> > I appreciate the authors' thorough response and the effort they put into the rebuttal. My concerns have been resolved, thus, I raise my score to Borderline accept.

---

> > > ### Author Response · Authors · 2025-08-07
> > >
> > > We thank the reviewer for pointing out key experiments which will strengthen the presentation of our work. We appreciate that the rebuttal satisfied the reviewer's concerns.

---

### Official Review · Reviewer_BfTw · 2025-07-02

**Clarity:** 4
**Significance:** 3
**Originality:** 3
**Rating:** 5
**Confidence:** 3

**Summary:**

This paper presents a thorough scaling exploration of AR image generation models based on discrete tokens. The authors highlight the trade-off between the reconstruction performance of autoencoders and the generation performance of AR models. They further demonstrate that increasing computational scale can help overcome this trade-off and push the performance boundaries of the model. Additionally, the paper introduces Causally Regularized Tokenization, which incorporates a next-token prediction loss to instill an inductive bias into the tokenizer. This approach leads to improved generation performance.

**Questions:**

Please see the weaknesses.

**Ethical Concerns:**

["NO or VERY MINOR ethics concerns only"]

**Final Justification:**

The authors address my concerns by:

- Committing to include qualitative visual comparisons in the final revision, explaining that larger rFID gaps cause more visible degradation, especially in small high-frequency regions.
- Conducting additional large-scale T2I experiments, demonstrating gains in gFID and CLIPScore.

Therefore, I keep my initial rating.

**Limitations:**

Yes.

**Paper Formatting Concerns:**

No formatting concerns.

**Quality:**

4

**Strengths And Weaknesses:**

**Strengths**

- This paper provides extensive experiments demonstrating that: (1) a trade-off exists between reconstruction (or compression) performance and generation quality; (2) scaling compute helps push the model's limits by analyzing this trade-off through the lens of scaling laws; and (3) incorporating an autoregressive inductive bias improves generation performance.
- These experimental results offer valuable insights for designing AR-based generative models.
- The proposed CRT and its optimized version are intuitively sound and empirically effective.
- The paper is well-written, logically structured, and supported by clear figures.

**Weaknesses**

- The authors could provide a qualitative comparison of the reconstructed results between the baseline VQGAN, CRT, and CRT with CE loss, as the rFID differences are quite subtle and may not translate into noticeable visual differences.
- It would be better if the authors could conduct some T2I generation experiments, which would make the work more generalizable.

---

> ### Author Rebuttal · Authors · 2025-07-28
>
> We thank the reviewer for their insightful review. We are glad that the reviewer appreciates our extensive experiments and writing. Below we respond to constructive feedback.
>
> **The authors could provide a qualitative comparison of the reconstructed results between the baseline VQGAN, CRT, and CRT with CE loss, as the rFID differences are quite subtle and may not translate into noticeable visual differences.**
>
> We thank the reviewer for this suggestion. Given the limitations around sharing visual media during the rebuttal period, we commit to showing these comparisons in the final revision. Qualitatively, the difference between reconstruction performance of VQGAN and VQGAN + CRT is hard to discern visually, and this is reflected in their rFID (2.21 vs 2.36 with CRT). We see this as an advantage of our method, as it mitigates damage to visual reconstrction while improving scaling. For larger gaps, e.g. 2.2 to 2.6 (VQGAN + CE CRT), we see a greater degradation that becomes more noticeable. This is prevalent in small high frequency patches (e.g. grass or tree branches) and in the bottom right corner of images (as the AR bias is applied in a raster-scan).
>
> **It would be better if the authors could conduct some T2I generation experiments, which would make the work more generalizable.**
>
> We thank the reviewer for pointing this out. To alleviate this concern and supplement our results, we ran a text-to-image experiment. For this experiment, we took the BLIP-3o long-caption [1] pre-training dataset (27M image-text pairs vs 1.2M ImageNet images), tokenized the text with OpenAI's CLIP L/14 text tower [2], and trained a auto-regressive image generation model conditioned on these text embeddings. This dataset is diverse, being a combination of webcrawled images and high-quality data from JourneyDB. We trained our XL sized model (775M) for 600k iterations at batch size 256. The results from this experiment are in the table below:
>
> | Tokenizer | gFID | CLIPScore |
> | --------------- | --------------- | --------------- |
> | Baseline | 4.61 | 0.33 |
> | CRT (ours) | 4.15 | 0.36 |
>
>
> We use CLIPScore [3] (higher is better) with OpenAI CLIP B/32 to measure image-text alignment. We see a solid improvement with our method, even though we are re-using the image tokenizer from our ImageNet experiments and this dataset is thus OOD. We will include full details of this experiment in the final revision.
>
> [1] BLIP3-o: A Family of Fully Open Unified Multimodal Models-Architecture, Training and Dataset, Chen et al 2025
>
> [2] Learning Transferable Visual Models From Natural Language Supervision, Radford et al 2021
>
> [3] CLIPScore: A Reference-free Evaluation Metric for Image Captioning, Hessel et al. 2021

---

### Official Review · Reviewer_DoZ7 · 2025-07-02

**Clarity:** 3
**Significance:** 2
**Originality:** 2
**Rating:** 5
**Confidence:** 4

**Summary:**

This paper studies the fundamental trade-off between image compression (stage 1) and generative modeling (stage 2) in two-stage image generation pipelines. Specifically, it investigates how the design of the tokenizer—namely the number of tokens and the codebook size—interacts with compute scaling laws. Through comprehensive empirical analysis, the authors find that under limited compute, high compression (i.e., fewer tokens and smaller codebooks) is beneficial, as it reduces reconstruction quality (rFID) but improves generative performance (gFID). To better navigate this trade-off, the authors introduce Causally Regularized Tokenization (CRT), a simple yet effective method that incorporates the autoregressive inductive bias of stage-2 decoders into the stage-1 tokenizer. CRT achieves significant efficiency gains (2–3×) and matches the performance of LlamaGen-3B with far fewer tokens and reduced model complexity.

**Questions:**

1. The paper states (line 158) that models on the training compute Pareto frontier of validation loss are also on the gFID Pareto frontier. How can this be concluded directly from Figure 4? Can you elaborate?

2. In line 220, you mention reducing training epochs by two to control training FLOPs. Was model convergence affected in those cases?

3. In Figure 7, CRT's rFID appears to be 2.36, but why the fitted curve has an intercept of 2.21?

4. CRT relies on the autoregressive modeling bias to guide tokenizer regularization. For other modeling paradigms (e.g., MaskGIT, diffusion), what inductive biases would be appropriate? Would CRT need to be redesigned accordingly? Any insights would be appreciated.

**Ethical Concerns:**

["NO or VERY MINOR ethics concerns only"]

**Final Justification:**

After reading the rebuttal and discussion, I have increased my score to a weak accept (5). The authors have adequately addressed my concerns regarding the scope of tokenizer choices, the effect of CRT near saturation, and the connection between the scaling analysis and the proposed method. The paper offers valuable empirical insights and a practical method (CRT) with strong efficiency gains. While some writing and structural issues remain—particularly the somewhat loose connection between the first half (scaling analysis) and the second half (CRT proposal)—the contributions are solid and of interest to the community.

**Limitations:**

yes

**Paper Formatting Concerns:**

No Paper Formatting Concerns.

**Quality:**

3

**Strengths And Weaknesses:**

**Strengths:**
1. The paper provides lots of evidence of how stage-1 tokenization choices (number of tokens and codebook size) affect stage-2 modeling performance, and how these trade-offs evolve under different compute budgets.

2. The proposed CRT mechanism is conceptually straightforward but effective, leveraging the AR modeling bias to regularize tokenizer outputs for better downstream performance.

3. The authors conduct targeted ablations to isolate the effect of key CRT components (e.g., applying the L2 loss before quantization), offering actionable insights into its design.

**Weakness:**

1. The analysis is limited to a fixed tokenizer architecture (VQ-GAN) and autoregressive stage-2 decoders. Interestingly, the experiments suggest that the specific choice of token count and codebook size does not strongly affect the compute-scaling behavior. This raises the question of whether other, potentially more impactful factors—such as the type of tokenizer or the choice of decoder architecture—might play a larger role in shaping the trade-off. However, this dimension remains unexplored.

2. CRT explicitly sacrifices reconstruction quality for better modeling ease. However, existing scaling laws suggest that rFID becomes a bottleneck at large model scales. Since the experiments are limited to models up to 800M parameters, it remains unclear whether CRT continues to help, plateaus, or even hurts performance in the high-capacity regime.

3. The first half of the paper focuses on empirical analysis of scaling laws and tokenizer design, while the second half introduces CRT as a solution. However, the connection between these two parts feels somewhat loose—does the first half merely identify a design choice (fewer tokens), or does it provide deeper insight that motivates CRT?

---

> ### Author Rebuttal · Authors · 2025-07-28
>
> ### Addressing Weaknesses
>
> **The analysis is limited to a fixed tokenizer architecture (VQ-GAN) and autoregressive stage-2 decoders.**
>
> We focused on VQGAN as it is the current SOTA tokenizer architecture for auto-regressive image generation. We note that the VQGAN architecture is still used for top discrete token models and has been dominant since it was proposed in \[1\]. We agree that tokenizer architecture would be an interesting avenue to study, however given the expense and depth of our scaling studies, we leave this axis of study to future work. We hypothesize more flexible (e.g. transformer-based) architectures can potentially gain more from causal regularization.
>
> **CRT explicitly sacrifices reconstruction quality for better modeling ease...  it remains unclear whether CRT continues to help, plateaus, or even hurts performance in the high-capacity regime**
>
> We agree that very close to saturation, reconstruction performance becomes important. We point out the following:
>
> 1. CRT only harms reconstruction performance minimally (2.21 vs 2.36 rFID), which is hard to differentiate visually.
> 2. rFID is not always the lower bound of generation model performance, although it usually serves as a good proxy. In scaling laws, $L_{min}$ (the scaling law's lower bound), $\alpha$ (the scaling exponent), and $\lambda$ (the intercept) are fit together [2, 3]. In our case, when we fit $L_{min}$ for gFID curves it generally matched rFID, so we used the convention of rFID as $L_{min}$. In two cases this was not true: 1. In Figure 6 (far left, 1k codebook size) and 2. In the CRT scaling law. In Figure 6 (far left) we used the fitted $L_{min}$ (2.7 instead of 3.0 rFID), and in the CRT scaling law, the fitted $L_{min} = 2.2$ which essentially matches the baseline tokenizer's rFID. Therefore, scaling laws suggest that CRT and the baseline tokenizer have the same saturation point. We will make this clear in the final revision.
> 3. We introduce the $CRT_{opt}$ recipe in the paper for the very purpose of optimizing for both absolute reconstruction performance and causal regularization. This improves performance near saturation (2.18 vs 2.35 gFID).
>
>
> **However, the connection between \[scaling laws and CRT\] does the first half merely identify a design choice (fewer tokens), or does it provide deeper insight that motivates CRT?**
>
> We thank the reviewer for this constructive observation. The first half of the paper shows that increasing bits per pixel above 16k codebook size and 256 tokens per image generally makes scaling worse. Further, while reducing to 1k codebook size improves the scaling law marginally, _explicitly_ reducing the reconstruction capacity to such a degree greatly harms generation performance close to saturation. This motivates us to look for a rate-distortion trade-off which does not directly inhibit the bottleneck size, of which CRT is one. Section 4 analysis expands on this concept. We will make this explicit in the text, which should improve the overall flow of the paper.
>
>
> ### Addressing Questions
>
> **The paper states (line 158) that models on the training compute Pareto frontier of validation loss are also on the gFID Pareto frontier. How can this be concluded directly from Figure 4? Can you elaborate?**
>
> This fact can be concluded directly from the figure, however it is difficult and we will improve the clarity in the final revision. To see this in the figure itself, each line is a specific parameter scale of increasing compute, so points in the validation loss curve directly correspond to points in the gFID curve. The pareto frontier are minimum points at a specific compute interval. For example, you see that in the validation loss curve, the last 6 points of the 775M parameter model lie on the pareto frontier. In the gFID curve this is the same for that model scale. To make this clear in the final revision, we will use stars to denote validation loss points which are on the pareto frontier and propagate those to the gFID curve.
>
> **In line 220, you mention reducing training epochs by two to control training FLOPs. Was model convergence affected in those cases?**
>
> Model convergence was not affected to any real degree. We trained versions at 38 and 40 epochs, and their rFIDs were 2.36 and 2.34 respectively, which we is within training noise.
>
> **In Figure 7, CRT's rFID appears to be 2.36, but why the fitted curve has an intercept of 2.21?**
>
> We expand on this above (rebuttal point 2 with regards to weakness point 2). Note that varying $L_{min}$ between 2.2 and 2.3 for CRT does _not_ change the scaling exponent, so our conclusions remain the same. We will make this nuance (fitting $L_{min}$ vs rFID lower bound) clear in the text.
>
> **CRT relies on the autoregressive modeling bias to guide tokenizer regularization. For other modeling paradigms (e.g., MaskGIT, diffusion), what inductive biases would be appropriate? Would CRT need to be redesigned accordingly? Any insights would be appreciated.**
>
> Yes, CRT would need to be re-designed if the stage 2 modeling design was changed. This is an active area of research for us. Looking towards diffusion, generally the closer the latent distribution is to a gaussian normal, the easier it is to do flow matching (since all the marginal flows cancel and you are left with no flow). However, if the latent prior is completely normal, and there is no flow, then only VAE decoder matters for generation. Further, the latent space would have a very low signal to noise ratio and the reconstruction would be blurry. This is one of the fundamental trade-offs for diffusion. This trade-off partially already exists in literature with the KL regularization term for the VAE prior, which pushes the latent distribution towards an isotropic normal, but there are other ways to accomplish this. For example, linearly interpolating the latent with noise during auto-encoder training accomplishes the same thing while allowing for a more controllable signal-to-noise ratio. Ultimately, for any stage 2 model, to introduce such an inductive bias into the stage 1 model requires analyzing the specific processes and assumptions inherent to stage 2 sampling.
>
> [1] Taming Transformers for High-Resolution Image Synthesis, Esser et al 2020.
>
> [2] Training Compute-Optimal Large Language Models, Hoffman et al 2022.
>
> [3] Scaling Laws for Neural Language Models, Kaplan et al 2020.

---

> > ### Comment · Reviewer_DoZ7 · 2025-08-05
> >
> > Thank you for your detailed response and clarification. Most of my concerns have been addressed, and I am happy to raise my rating to 5 (Accept).

---

> > > ### Author Response · Authors · 2025-08-07
> > >
> > > We appreciate the response and are pleased that all concerns have been addressed. We further thank the reviewer for noting where we can improve the paper for clarity and believe this discussion has strengthened our final revision.

---

### Official Review · Reviewer_SM8A · 2025-07-02

**Clarity:** 4
**Significance:** 4
**Originality:** 4
**Rating:** 6
**Confidence:** 4

**Summary:**

The submission has two main contributions.

The first is a thorough study on the design space of discrete image tokenizers (stage 1), and what effect it has on downstream generative modeling with vanilla autoregressive models (stage 2). To that end, the paper sets up scaling laws and studies the relation of tokenizer vocabulary size and token sequence length with stage 2 model scale & compute.

The second main contribution is a training modification on 2D tokenizers that regularizes the learned latent space to be more amenable to generative modeling with AR models. The authors call this "Causally Regularized Tokenization" (CRT) and it is performed by co-training a tiny AR model with the tokenizer whose objective it is to perform next-token prediction on the tokenizer latents. By doing that, the tokens are trained to be more predictable in an AR manner. The authors observe a slight reduction in reconstruction performance, but at the benefit of significantly stronger generative performance, with better scaling behavior compared to the non-CRT baseline.

**Questions:**

- The authors note that raster-scan order has been observed to work best for AR generation, but how would the use of CRT change this? Is this simply an inductive bias that may be arbitrarily changed by regularizing the tokenizer to encode information in a different order?
- Fig 3: It may not be practical for stage 2 training, but it would be interesting to see the trendlines for scaling the codebook size extended beyond 2^17. It looks like for PSNR, scaling the codebook size may scale better.

**Ethical Concerns:**

["NO or VERY MINOR ethics concerns only"]

**Final Justification:**

My initial review for this submission was positive and the author's rebuttal addressed most of my concerns, and even provided additional insights (especially with respect to the VQ training recipe and the order in which CRT is applied) which could further strengthen the paper. My main issues with the submission were that the main evaluation was on ImageNet generation, but the authors showed that CRT is effective for text-to-image generation on a more diverse and large-scale dataset too. For these reasons I will further increase my score.

**Limitations:**

The authors have adequately addressed potential limitations.

**Paper Formatting Concerns:**

- L54: The bullet points in the summary of contributions are too compact
- Nit: The current Fig. 3 placement is somewhat far from the section that discusses it

**Quality:**

3

**Strengths And Weaknesses:**

Strengths:
- The investigations into how a tokenizer's codebook size and token sequence length should be chosen for different sizes and compute budgets of downstream generative models are highly interesting and well-designed.
- The design of CRT is simple and effective, and incurs a minimal compute overhead. The efficiency improvements / improved scaling law slope, and the demonstration of outperforming LlamaGen with smaller models and shorter sequence lengths are impressive.
- The authors seem to have put great effort to taking into account the compute overhead of various interventions and to compute-match the various experiments. In general, the experimental setup and procedure appears very thorough and thought-through.
- The ablations on CRT are thorough and answered all the key questions I had w.r.t. loss balancing, size of the AR regularizer, loss function, etc.
- The paper shows sufficient training details to be able to reproduce the experiments and method
- The paper's writing is very clear, the sections are well-formatted, and the figures are easy to parse. The paper is packed with interesting and useful insights, and it was a joy to read. It also challenges some common assumptions from previous works, which would be useful to study in more detail.

Weaknesses:
- Sec 3.4: The authors note that more compressed sequences are generally more compute optimal, but how much of this may be due to the tokenizer operating on out-of-distribution resolutions? It is noted that for these experiments, the same tokenizer is used, but at higher resolutions than what it was trained at. Would the results change if tokenizers were trained at those resolutions? Would we see further improvements for smaller models if the sequence length is further decreased, e.g. to 14*14 or below (with frozen or specially trained tokenizers)?
- The paper focuses mostly on class-conditional ImageNet generation, which to me is one of its main weaknesses. There are some results on LSUN, but as for IN1K generation, the benchmark is somewhat close to saturation and susceptible to gaming if one tries to just overfit a model to the train set (since gFID is measured against it). The paper's findings are valuable, but could be significantly stronger if they were demonstrated to A) hold on more difficult datasets, B) on more nuanced tasks like text-to-image where the conditioning alignment could be measured, and C) in a data regime that is not data-limited, unlike ImageNet-1k.
- Besides the main contributions, the VQ training recipe proposed by the authors seems to outperform simpler baselines like FSQ. This is somewhat surprising, as the VQ recipe in this paper seems similar to the one studied in FSQ, and it would be useful to investigate this further and show if the proposed CRT also works well for LFQ.
- The paper could benefit from including a discussion about related work like JetFormer [1] that jointly trains a tokenizer with an AR model, as well as alternative methods of regularizing tokenizers with a 1D causal structure, like SEED [2], ElasticTok [3], One-D-Piece [4], FlexTok [5], or Semanticist [6]. I note that the submission already includes a discussion about LARP's use of AR regularization.

[1] JetFormer: An Autoregressive Generative Model of Raw Images and Text, Tschannen et al. 2024

[2] Planting a SEED of Vision in Large Language Model, Ge et al. 2023

[3] ElasticTok: Adaptive Tokenization for Image and Video, Yan et al. 2024

[4] One-D-Piece: Image Tokenizer Meets Quality-Controllable Compression, Miwa et al. 2025

[5] FlexTok: Resampling Images into 1D Token Sequences of Flexible Length, Bachmann et al. 2025

[6] "Principal Components" Enable A New Language of Images, Wen et al. 2025

---

> ### Author Rebuttal · Authors · 2025-07-28
>
> We thank the reviewer for their in detailed and insightful review. We were particularly gratified that the reviewer found our paper "a joy to read" and that "the experimental setup and procedure appears very thorough and thought-through". We address the reviewers constructive feedback/criticisms below.
>
> ### Addressing Weaknesses
>
> **The authors note that more compressed sequences are generally more compute optimal, but how much of this may be due to the tokenizer operating on out-of-distribution resolutions?**
>
> We thank the reviewer for pointing this out! In order to address the point of operating OOD resolutions, we have results for:
> - _Tokenizers greater than 16x16 resolution_: We do not see significant improvement in rFID/other distortion metrics when training with natively higher input resolution, nor do we see stage 2 improvements. We ran these experiments with a 211M parameter stage 2 model for 500k iterations (batch size 256).
> | Tokens per Image | rFID / gFID | Native resolution training |
> |------------------|-------------|----------------------------|
> | 400 | 1.30 / 3.76 | no |
> | 400 | 1.21 / 3.72 | yes |
> | 576 | 0.99 / 3.61 | no |
> | 576 | 0.93 / 3.66 | yes |
> - _Tokenizers less than 16x16 resolution_: It is likely possible to see some improvement since training a lower resolution tokenizer often damages PSNR; however, we note that training such a tokenizer requires architectural modifications in order to avoid resizing the image below 256x256. This introduces an architectural confound outside the scope of this paper. We will include this comparison when the model is finished training. We thank the reviewer for their great observation.
>
>
> **The paper focuses mostly on class-conditional ImageNet generation ... The paper's findings are valuable, but could be significantly stronger if they were demonstrated to A) hold on more difficult datasets, B) on more nuanced tasks like text-to-image where the conditioning alignment could be measured, and C) in a data regime that is not data-limited, unlike ImageNet-1k.**
>
> We thank the reviewer for pointing this out. To alleviate this concern and supplement our results, we ran a text-to-image experiment. For this experiment, we took the BLIP-3o long-caption [1] pre-training dataset (27M image-text pairs vs 1.2M ImageNet images), tokenized the text with OpenAI's CLIP L/14 text tower [2], and trained a auto-regressive image generation model conditioned on these text embeddings. This dataset is diverse, being a combination of webcrawled images and high-quality data from JourneyDB. We trained our XL sized model (775M) for 600k iterations at batch size 256. The results from this experiment are in the table below:
>
> | Tokenizer | gFID | CLIPScore |
> | --------------- | --------------- | --------------- |
> | Baseline | 4.61 | 0.33 |
> | CRT (ours) | 4.15 | 0.36 |
>
>
> We use CLIPScore [3] (higher is better) with OpenAI CLIP B/32 to measure image-text alignment. We see a solid improvement with our method, even though we are re-using the image tokenizer from our ImageNet experiments and this dataset is thus OOD. We will include full details of this experiment in the final revision.
>
>
> **Besides the main contributions, the VQ training recipe proposed by the authors seems to outperform simpler baselines like FSQ ... it would be useful to investigate this further and show if the proposed CRT also works well for LFQ.**
>
> This point has two parts, exploring the VQ recipe and discussion of LFQ.
>
> - **VQ recipe exploration.** The performance of the VQ recipe was also surprising to us. Our VQ recipe is similar to that of ViT-VQGAN and LlamaGen but ended up being more scalable than both. We credit this to a long learning rate warmup period, which is the only substantial deviation from the ViT-VQGAN/LlamaGen settings. [4] finds that long learning rate warmup periods lend greater stability through training runs even when instabilities appear far after the warmup period. It's possible this has a similar effect here. Not only is our codebook utilization ~100%, Figure 9 (center left) implies that the distribution of code usage is close to uniform _without any entropy losses_. This is a deviation from the result in FSQ, where both FSQ and VQ are substantially below the uniform compression cost threshold (see Figure 3 bottom right in FSQ). We show a the effect with and without learning rate warmup below for our setting, where we see a clear improvement with warmup across codebook sizes:
>
>
> | Codebook Size | With Warmup (rFID) | Without Warmup (rFID) |
> | --------------- | --------------- | --------------- |
> | $2^{12}$ | **2.60** | 3.04 |
> | $2^{13}$ | **2.53** | 2.95 |
> | $2^{14}$ | **2.21** | 2.24 |
> | $2^{15}$ | **1.91** | 2.29 |
> | $2^{16}$ | **1.76** | 2.45 |
>
>
>  - **Regarding LFQ.** We have started training a tokenizer with LFQ, but properly training LFQ requires carefully tuning entropy loss and commitment loss correctly, and therefore it is not finished as of the rebuttal deadline. We will include this comparison when it is complete. We speculate that it will not outperform our VQ recipe in terms of rFID, as we are already achieving almost uniform codebook utilization (Figure 9).
>
> **The paper could benefit from a discussion of JetFormer and more 1D tokenization works.**
>
> We thank the reviewer for pointing out these important works. We will include a discussion of them in the final revision under the 1D tokenizer section in our related work.
>
>
> ### Addressing Questions
>
> **The authors note that raster-scan order has been observed to work best for AR generation, but how would the use of CRT change this?**
>
> We have done experiments with alternate token orders (Hilbert space filling curve, spiral in, spiral out, snake order from top left) at small scale (<200M stage 2 parameters). In all of these experiments, raster scan ended up being the best empirically, often substantially (>1 gFID). When applying CRT to other orderings (at small scale), CRT would improve performance but not enough to overcome the absolute difference between raster-scan and worse orderings. We are not the first to note the superiority of raster-scan. Indeed a comparison exists in [5] in Figure 47 (although with a sliding window attention pattern instead of full causal). It is possible that a more flexible base architecture than that of VQGAN would allow for more flexible re-ordering. However our implementations of ViT-VQGAN underperformed VQGAN when compute-matched, so we focused on VQGAN for this study.
>
>
> **Fig 3: It may not be practical for stage 2 training, but it would be interesting to see the trendlines for scaling the codebook size extended beyond 2^17. It looks like for PSNR, scaling the codebook size may scale better.**
>
> This is an interesting observation. In response, we trained a tokenizer with $2^{18}$ codes. Note that if we trained a stage 2 model, the embedding parameters for a 770M GPT model would be ~650M parameters, which as the reviewer notes is cumbersome. We actually do see a slight improvement over increasing tokens per image (rFID: 1.72, PSNR: 21.6, MS-SSIM: 0.695), however it's not great enough to justify the added compute. To stabilize this run during the GAN loss phase, we had to introduce warmup when increasing the weight of the discriminator loss (over 2000 iterations). We did not have to do this for codebook size 131k.
>
> \[1\] BLIP3-o: A Family of Fully Open Unified Multimodal Models-Architecture, Training and Dataset, Chen et al 2025
>
> \[2\] Learning Transferable Visual Models From Natural Language Supervision, Radford et al 2021
>
> \[3\] CLIPScore: A Reference-free Evaluation Metric for Image Captioning, Hessel et al 2021
>
> [4] Small-scale proxies for large-scale Transformer training instabilities, Wortsman et al 2023
>
> [5] Taming Transformers for High-Resolution Image Synthesis, Esser et al 2020.

---

> > ### Comment · Reviewer_SM8A · 2025-08-05
> >
> > I thank the authors for their rebuttal, which addressed most of my concerns and questions. I think that the text-to-image results and insights into the effects of warmup for VQ training are especially interesting and worth including in the submission. I would also recommend to include the study of the choice of token order, since it seems in line with previous findings and the insights that CRT improves all variants but cannot overcome the choice of ordering are useful to better understand what the CRT regularization does. I currently do not have other major concerns about the submission.

---

> > > ### Author Response · Authors · 2025-08-07
> > >
> > > We are glad that our rebuttal addressed the reviewer's concerns. We particularly appreciate the detail and care put into this review and appreciate the constructive criticisms, which we believe will enhance our paper.

---

> ### Comment · Area_Chair_JLwk · 2025-08-05
>
> Dear Reviewer SM8A,
>
> As we're approaching the end of author-reviewer discussion period, please read the rebuttal and start discussion with the authors as soon as possible. If all your concerns have been addressed, please do tell them so. Please note that submitting mandatory acknowledgement without posting a single sentence to authors in discussions is not permitted. Please also note that __non-participating reviewers will receive possible penalties of this year's responsible reviewing initiative and future reviewing invitations.__
>
> Thanks,
>
> AC

---

### Comment · Area_Chair_JLwk · 2025-08-03

Dear Reviewers,

Thanks for your hard work during the review process. We are now in the author-reviewer discussion period.

Please (1) carefully read all other reviews and the author responses; (2) start discussion with authors if you still have concerns as early as possible so that authors could have enough time to response; (3) acknowledge and update your final rating. Your engagement in the period is crucial for ACs to make the final recommendation.

Thanks,

AC

---

### Decision · Program_Chairs · 2025-09-17

**Decision:**

Accept (spotlight)

**Comment:**

This paper studies the trade-off between image tokenization and generative modeling in two-stage image generation methods, and proposes a regularization term for tokenization to benefit the training of the second stage. Reviewers acknowledged the major contribution and valuable insights of the proposed method, while initially raising concerns regarding its experiment designs and missing T2I results.

After the rebuttal, the authors addressed most of the concerns, and all reviewers agreed to accept this paper. AC read all the reviews, author rebuttals, and the paper, and believes this is a strong paper and recommends acceptance.